# Enhancing the Catalytic Performance of Zeolites via Metal Doping and Porosity Control

**DOI:** 10.3390/molecules30183798

**Published:** 2025-09-18

**Authors:** Linda Zh. Nikoshvili, Lyudmila M. Bronstein, Valentina G. Matveeva, Mikhail G. Sulman

**Affiliations:** 1Department of Biotechnology, Chemistry and Standardization, Tver State Technical University, A.Nikitina Str., 22, 170026 Tver, Russia; 2Department of Chemistry, Indiana University, 800 E. Kirkwood Av., Bloomington, IN 47405, USA

**Keywords:** hierarchical zeolites, metal-doped zeolites, chemical etching, template methods, catalytic pyrolysis, alcohols to hydrocarbons, cracking, hydrocracking, hydroisomerization, hydrodeoxygenation

## Abstract

Zeolites are widely used as solid acid catalysts and also as supports in complex multifunctional heterogeneous systems. In recent years, there has been an increase in the development of zeolite-based catalysts with hierarchical porosity combined with metal dopants (modifiers or catalysts). These modifications can significantly improve the catalytic characteristics of such materials. In this review, we discuss the application of hierarchically porous zeolites, including metal-doped ones, in catalytic reactions employed in the production and upgrading of liquid fuels, i.e., pyrolysis of biomass and polymeric wastes; conversion of alcohols to fuel hydrocarbons, aromatics and olefins; cracking and hydrocracking of polymeric wastes and hydrocarbons; and hydroisomerization. It is revealed that, in many cases, higher activity, selectivity and stability can be achieved for metal-doped hierarchical zeolites in comparison with parent ones due to control over the diffusion, surface acidity and coke deposition processes.

## 1. Introduction

Zeolites are widely used as porous solid acid catalysts and as supports for catalytically active metal-containing species. Typically, zeolites consist of subnanometer interconnected channels that cause a number of issues due to strong diffusion limitations for many reactants [1,2,3] and fast catalyst deactivation due to coke formation [4,5].

Metals can be introduced into the catalyst’s composition via several methods:(i)A one-pot method, when a metal precursor is added to the mixture of reagents used for the zeolite synthesis [6,7,8]. In this case, metal ions are incorporated into the zeolite framework.(ii)Anion exchange with or without further reduction leading to the formation of metal clusters [9,10,11].(iii)Incipient wetness impregnation [6] leading to the formation of nanoparticles.

In the case of the impregnation method, metal-containing particles are mainly located on the zeolite outer surface. Such particles are accessible to the reagents; however, the resulting catalytic systems can display low stability. When metal particles are encapsulated or fixed (Figure 1) in zeolite pores, they become resistant to agglomeration and leaching [9], but in this case, the effect of molecular sieving of micropores will have a strong influence on mass transfer.

In the last decade, there has been constantly growing interest in zeolites containing meso- and macropores, including zeolites with hierarchical porosity [2,5], as well as zeolites whose structures are doped with different metals [10,12]. On the one hand, zeolites with hierarchical porosity allow one to encapsulate catalytic metal nanoparticles into the mesopores and preserve them from leaching and aggregation [13]. On the other hand, the presence of a share of microporous regions in hierarchical zeolites allows for molecule separation effects and, in general, makes the zeolite framework more stable.

At present, hierarchical zeolites are rarely used at industrial scale due to costly methods of preparation and some environmental issues [14,15]. Hierarchical titanium-silicate-1 (TS-1), as well as the mesoporous zeolites mordenite and Y [5] (the latter obtained by the surfactant template post-treatment), represent examples of successful industrial applications. We believe that the implementation of hierarchical zeolites in industry will rise continuously due to the development of one-pot simplified technologies using environmentally friendly reagents for the preparation of hierarchical zeolites. In this review, we analyzed recent accomplishments (made in the last ten years) in the field of catalytic materials based on hierarchical zeolites, including metal-doped materials.

Currently existing methods of the synthesis of zeolites with hierarchical porosity can be divided into top-down and bottom-up approaches [2,4,5,14,16]. These approaches are described in detail in a series of reviews by Hartmann et al. [5], Feliczak-Guzik et al. [2], Kerstens et al. [14] and Roth et al. [16]. Below, we will provide a brief overview of the most popular methods.

The group of top-down methods is based on the treatment of zeolites with different reagents (e.g., hydroxides and salts of alkaline metals and ammonium—NaOH, KOH, NH_4_OH and NH_4_F—and some acids—HCl, HF, HNO_3_ and oxalic acid) in order to provide a partial dissolution of Si, in the case of alkaline treatment, or Al, in the case of acidic treatment, in the zeolite crystal lattice. The chemical treatment of zeolites is the simplest, cheapest and most widespread approach, but it also has several drawbacks. It is noteworthy that the acid treatment results in the dissolution of amorphous zeolite fragments [17]. Acid treatment changes zeolite acidity and reduces coke formation during catalytic reactions. However, the blockage of zeolite pores because of the redeposition of weakly acidic dissolved species can also take place, which may have both positive and negative impacts on the catalyst performance [17]. In the case of desilication, the volume of micropores decreases more significantly (Figure 2), as compared to dealumination [18], especially for zeolites with a high silica content. During desilication, the reinsertion of aluminum atoms into the walls of mesopores can also take place [17].

At present, the most promising strategy is to combine dealumination and desilication [19] to create mesoporosity and, at the same time, to remove aluminum-rich species—the reason for coke formation. For example, Qin et al. [20] proposed consecutive treatment of a MOR-type zeolite with oxalic acid (designated as MOR-A) and NH_4_F (designated as MOR-A-AF). It was shown that the first stage of the treatment with the acid resulted in the formation of framework defects, which further (at the second stage, i.e., NH_4_F etching) favored the dissolution of the zeolite framework and the formation of mesopores (Figure 3) [20]. In the dealkylation reaction of 1,3,5-triisopropylbenzene, the modified zeolite MOR-A-AF produced a more than six-times higher conversion (67%) as compared to MOR-A. The content of Al was three-times lower in MOR-A-AF than in the parent MOR [20].

During the chemical etching, the process of pore formation can be governed by the so-called pore-directing agents—small organic molecules, which are able to provide “interior protection” of framework and subsequent zoned dissolution within the crystals (Figure 4) [21]. Thus, the resulting zeolites possess not only hierarchical porosity but also high crystallinity and preserved microposity, providing higher stability of such zeolites in comparison with meso/macroporous samples.

In some cases, a chemical treatment can be combined with the use of surfactants [22], which play the role of structure-directing agents. The treatment can be carried out under hydrothermal conditions [2,5,19,22,23], leading to subsequent condensation of dissolved species and the formation of secondary porosity. For example, Suarez et al. [22] synthesized hierarchically porous zeolite through the hydrothermal treatment of commercial zeolite Beta with cetyltrimethylammonium bromide (CTAB) in the presence of NaOH or NH_4_OH for further application in the isomerization/disproportionation of *m*-xylene (a model reaction). The treatment was carried out for 6–12 h at 80 °C (in case of NaOH) or 150 °C (in case of NH_4_OH). This treatment allowed the formation of ~3 nm mesopores, while the micropore volume was decreased. These events were confirmed by the bimolecular disproportionation reaction of *m*-xylene.

The bottom-up approach is based on the use of various removable templates, such as organic compounds, including polymers [12,24,25,26,27,28], carbon [3,29,30,31], and calcium carbonate [6,32]. The recent examples show that carbonaceous templates can have a highly ordered porous structure [3,31]. Liu et al. [31] applied the Ordered Macro-Mesoporous Carbon (OMMC) as a template for the synthesis of ZSM-5 single crystals with tunable porosity (Figure 5). The acidity of hierarchical ZSM-5 can also be adjusted via changing the molar ratio of SiO_2_/Al_2_O_3_ during the zeolite synthesis. The material developed allowed high conversion of benzyl alcohol (about 90%) and nearly 33% selectivity to the alkylation product. These values of conversion and selectivity were nearly double compared to those of commercial ZSM-5 due to the optimal acidity and boosted diffusivity of hierarchical ZSM-5 [31].

In a similar way, Zhang et al. [3] synthesized macroporous zeolite (OMbeta) via the bottom-up template approach (Figure 6). Ni_2_P particles supported on OMbeta were used for hydrodenitrogenation of quinoline and revealed higher catalytic activity, as compared to the traditional Ni_2_P/H-beta catalyst. The catalyst stability was not estimated in this study [3], but it is important to emphasize that macroporous systems typically possess lower thermal and hydrothermal stability, as compared to microporous zeolites (confirmed by both the experimental data and by the results of the computational study [33]). Recently, Gao et al. [34] reported successful synthesis of extra-stable material ZEO-5 consisting of extra-large pores with Ti species. This material was utilized in the production of propylene oxide, revealing a future promising route to catalytic materials obtained via the template method.

As an alternative approach, the regulation of kinetics of zeolite growth can result in the formation of hierarchical porosity [35]. This multistep procedure requires thorough control of the ripening processes via the addition of organic molecules (i.e., surfactants) or inorganic ions and the adjustment of a required temperature [2,35].

Recently, the single-step synthesis of hierarchical zeolites was proposed [1], which was based on the “impurity”-blocking effects. This means the introduction of metal heteroatoms, preventing the formation of a smooth surface during zeolite growth. For example, single crystal hierarchical zeolites were obtained using Ti species (as a stopper) with strong adsorption on the crystal surface, resulting in the fractal growth and the formation of surface protrusions (Figure 7) [1].

Both the top-down and bottom-up approaches can be used to synthesize pillared zeolite materials [36]. Such materials can be obtained either by one-stage direct synthesis or using preliminarily formed 2D-layered precursors [37]. “Pillaring” allows one to create materials for adsorption and catalysis with the enhanced diffusivity and stability, as compared to conventional 2D-layered solids. To date, about twenty of the existing zeolite frameworks can form 2D layers [16,37,38], among which zeolites of MWW [7,39,40,41,42,43,44] and MFI [8,11,44,45,46,47,48,49,50,51,52,53] families are the most popular. The authors of [16] indicated that layered MFI is of greater interest than MWW, since vertical pores across the MFI layers allow for additional paths of molecular diffusion [45]. A variety of methods for the synthesis of layered zeolite precursors as well as different structure-directing agents and templates have been reported [37]. Moreover, pillars can be obtained from different materials, such as surfactants, oxides (i.e., SiO_2_), and zeolites [39,44,53]. The one-step method for the synthesis of pillared nanosheets (referred as self-pillared pentasil (SPP) zeolites) in the presence of structure-directing agents was proposed for the MFI framework [45]. At present, the main issue impeding the wide catalytic application of pillared zeolites is the complexity and variety of conditions for the preparation of 2D structures [44]. Recently, an environmentally friendly synthesis of SPP without any seeds or templates was reported [11,49], which opens new possibilities for practical applications of such materials [14,15]. Shamzhy et al. [44] mentioned that the catalytic performance of 2D materials in comparison with their 3D analogues depends on the reaction. For example, 2D zeolites may possess enhanced activity in catalytic pyrolysis, while in the case of isomerization reactions, no significant effects were found [44]. It is remarkable that the pillared zeolites consisting of 2D layers can swell, allowing for confinement of metal nanoparticles into the zeolite cavities [39].

A general scheme of approaches for the synthesis of hierarchical zeolites is presented in Figure 8.

We would like to emphasize that this review analyzes recent works in the field of catalytic applications of zeolites with regulated porosity and metal doping. Among numerous reactions, which can be carried out in the presence of hierarchical zeolites (oxidation [54,55,56,57,58,59], epoxidation [60], reduction of nitrogen oxides [61,62,63] and organic compounds [64,65], condensation [66], methanation of carbon dioxide [67,68], methylation [69], etc.), we focus on cracking and hydrocracking; hydroisomerization; hydrodeoxygenation; aromatization; pyrolysis and bio-oil upgrading; alcohols to fuels, i.e., the reactions, which are relevant to syntheses and processing of liquid fuels. We examine the most striking examples, by which the benefits of control over the zeolite porosity and composition can be illustrated. Both metal-free and metal-containing zeolites are being considered.

## 2. Catalytic Application of Hierarchical Zeolites

### 2.1. Catalytic Pyrolysis

#### 2.1.1. Pyrolysis in the Presence of Metal-Free Zeolites

Catalytic pyrolysis is an important industrial technology, which allows for the processing of various biomass and plastic wastes into feedstock for organic syntheses, gaseous and liquid fuels, and solid sorbents [70,71]. The key component of catalytic pyrolysis is the solid catalyst. Among numerous pyrolysis catalysts, there are metal salts and oxides, carbons modified with acid groups, and zeolites [71,72]. Zeolites are the most widespread [71,72,73,74]. Regulating the acid/base properties of zeolites, their morphology, porosity and presence of metal dopants, it is possible to enhance their activity, selectivity, stability, and resistance to coke formation in catalytic reactions. The increase in catalyst stability is especially important in the case of the transformation of oxygen-rich feedstock with a high moisture content.

Catalytic pyrolysis can be carried out either *in situ* (with direct contact of the catalyst with biomass or polymer raw materials) or *ex situ*. In the latter case, there are two reactors or two working zones in which first thermal pyrolysis takes place followed by the second step, a catalytic reactor/fixed bed for the upgrading of pyrolysis vapors.

In the last decade, there were numerous examples of the use of metal-free hierarchical zeolites (mostly ZSM-5) in catalytic pyrolysis of plant biomass [75,76,77,78,79,80,81,82,83,84,85,86,87,88,89], synthetic polymers [90,91], and mixtures of biomass and polymeric wastes [92].

Gamliel et al. [75] synthesized a series of MFI zeolites using two approaches:(i)Bottom-up approach where the well-known template for the ZSM-5 synthesis, tetrapropylammonium hydroxide (TPAOH) [93], was used in combination with NaOH addition and hydrothermal treatment (the sample designated MFI-Meso);(ii)Top-down approach where desilication was carried out by the treatment with NaOH solutions of different concentrations (MFI-SA-Mild and MFI-SA-Strong) with the addition of a surfactant (CTAB) (the samples designated MFI-DS-Mild and MFI-DS-Strong).

All the zeolites were tested in the *in situ* catalytic fast pyrolysis (CFP) of cellulose and biomass of *Miscanthus* × *giganteus*. The highest yield of mono-aromatic hydrocarbons (MAHs) in the case of mesoporous zeolites as compared to the ones with predominant microporosity (MFI-100 nm and MFI-Pa) was likely due to the accessibility of acid sites. It was also found (Figure 9) that among the hierarchical zeolites, the yield of MAHs was highest for the catalysts with medium mesoporosity, i.e., for MFI-DS-Mild, which underwent mild desilication with 0.1 M NaOH at 65 °C for 30 min, while the product distribution was nearly the same for all the zeolites [75]. It was proposed [75] that, when mesoporosity increases, MAHs undergo polymerization that results in an increase in the yield of poly-cyclic aromatic hydrocarbons (PAHs) and coke formation (Figure 9). Thus, the mesopore volume along with surface acidity was crucial for the PAH formation, rather than the mesopore diameter. The negative impact of excessive mesoporosity was also reported by Zhang et al. [77], who synthesized a series of hierarchical HZSM-5 samples using the bottom-up approach with TPAOH and CTAB as micro- and mesopore-generating templates. The *in situ* CFP of the rice straw to aromatics revealed that the high volume of mesopores (≥68% of the sum of *V_micro_* and *V_meso_*) results in damage to the crystal structure and weakens the zeolite acidity, which, in turn, causes a lower aromatics yield and higher coke formation.

Regarding the coke formation, Jia et al. [76] emphasized that the type of coke forming during the CFP is one of the important parameters distinguishing hierarchical zeolites from the parent microporous systems. They reported [76] that the coke can be in three forms: (i) toxic coke located in micropores, (ii) coke precursors located in mesopores, and (iii) thermal coke formed from heavy molecules on the outer surface. The process of external coke deposition depends on the following factors: the rate of volatiles formation; mass transfer of volatiles to the outer surface of zeolite; reactivity of volatiles and the rate of their diffusion inside the zeolite. Thus, hierarchical zeolites have an advantage due to the enhanced mass transfer of volatiles (Figure 10).

In CFP of oak in a microfluidized bed reactor (MFBR) at 500 °C in the presence of hierarchical HZSM-5 (obtained by the alkaline desilication of parent zeolite [76]), MAHs and bi-aromatics were formed faster in the case of hierarchical zeolite in comparison with the parent one. Moreover, the parent zeolite deactivated faster than the hierarchical HZSM-5. At the same time, the formation of coke precursors inside the mesopores does not result in their blockage; the *V_meso_* started decreasing only at a biomass-to-catalyst ratio (BCR) higher than 0.85.

The superior stability of hierarchical zeolites to external coke deposition in contrast to the parent microporous materials was also demonstrated by Bi et al. [78], who studied the *in situ* CFP of kraft lignin in the presence of desilicated HZSM-5 and H-β.

An interesting comparison of three types of hierarchical zeolites (mordenite (MOR), beta (*BEA), and ZSM-5(MFI)) was carried out by Pinard et al. [82]. The selectivity toward aromatics and coke formation depended on the shape of zeolite channels. Among the chosen zeolites, hierarchical MFI provided the highest yield of MAHs and, at the same time, the lowest coke formation (Figure 11), likely due to better accessibility of Brønsted acid sites (BAS) and shape selectivity of MFI channels.

MFI after mild desilication provided the highest yield of MAHs (about 5 wt.% after the *ex situ* CFP of oak in a double fixed-bed microreactor at 773K). This is consistent with the earlier findings of Gamliel et al. [75] (see above). Moreover, in the case of MFI and *BEA (zeolites with highly interconnected 3D pores), deactivation was due to poisoning by coke rather than the pore blockage. At the same time, MOR (zeolite with 2D pores) underwent fast deactivation due to pore plugging despite hierarchical pores [82].

In contrast to the work of Pinard et al. [82], in which hierarchical *BEA obtained by the conventional chemical treatment was inferior to MFI in terms of the MAH yield [82], Wu et al. [83] showed the advantages of hierarchical Beta zeolites in comparison with ZSM-5 in the process of the *in situ* CFP of lignin at 600 °C. Hierarchical Beta zeolites were prepared by the treatment of parent Beta (B36), where 36 is the Si-to-Al ratio (SAR), with tetraethylammonium hydroxide (TEAOH) at 140 °C. The increase in *V_meso_* resulted in a corresponding decrease in the solid yield due to the better stabilization of bulky phenols and their subsequent conversion to gaseous and liquid products. However, the maximum yield of bio-oil (13.1 wt.%) was found for the sample (B36-0.3-24) with the medium mesoporous surface area and volume. The highest MAH selectivity (56.0 area%) was also found for hierarchical B36-0.3-24. It was proposed that 12-membered ring (12-MR) channels of B36-0.3-24 facilitated the fast diffusion and decomposition of pyrolysis intermediates, while its low acidity favored the formation of cyclopentadiene and aromatics. In contrast, the 10-MR channels of ZSM-5 (36) hindered the accessibility of bulky alkoxyphenols into inner acid sites and, hence, led to the low aromatics yield.

Recently, Butt et al. [84] synthesized defect-free nanocrystals (ultra-thin ZSM-5-F) for catalytic pyrolysis of stem wood and grot (in both cases using the *ex situ* and the *in situ* modes). In the *in situ* experiments, the ZSM-5-F catalyst provided selectivity to benzene, toluene, and xylene (BTX) of 20.6% that was three-fold greater compared to that of hierarchical HBeta. The high selectivity of ZSM-5-F to MAHs and the lower PAHs formation were attributed to the well-structured channels with the higher average pore size (12 nm), as compared to other samples studied with 6–8 nm pores. In addition, the defect-free framework of ZSM-5-F favored a reduction in mass transfer limitations. In the *ex situ* experiments, the ZSM-5-F catalyst resulted in 30 wt.% of the jet fuel formation while converting 93wt.% of the pyrolysis oil consisting of syringol, guaiacol, methanol (MeOH), and water [84].

Zhou et al. [89] reported the synthesis of hierarchical ZSM-5 by the addition of organosilanes (OSAs) to the zeolite precursor (Figure 12). OSAs with different alkyl chain groups were used to investigate the anchoring effect. It was shown that OSAs can modulate the coordination environment of Si in zeolites and, thus, promote the formation of acid sites. The ODDMMS (0.05)-Z5 zeolite with the highest Brønsted acidity and the larger external surface area and total pore volume produced the aromatics yield of 42.2% for the *in situ* CFP of cellulose at 600 °C [89].

As mentioned above, hierarchical zeolite catalysts can be also employed in the pyrolysis of polymeric wastes. For example, He at al. [90] studied the *ex situ* catalytic pyrolysis of polypropylene (PP) at 550 °C in the presence of HZSM-5 with different SARs (23, 50 or 80) treated with NaOH in the presence of CTAB. As could be expected, the samples with lower SARs were less prone to hydrolysis. The highest *V_meso_* was found for the hierarchical zeolite with the SAR of the parent HZSM-5 equal to 80. This hierarchical sample provided the best propylene yield (41 wt.%) and the yield of total light olefins and MAHs of 92 wt.%. It was emphasized that the increase in pore sizes in conjunction with maintaining strong acid sites favors propylene production. Qie et al. [91] reported the microwave (MW)-assisted chelation–alkaline(Na_2_H_2_EDTA-NaOH) treatment of ZSM-5. Variation in the concentration of a chelating agent led to a change in the pH value and, hence, the mechanism of secondary porosity formation (Figure 13).

The catalyst Z-HT-0.3-3 (0.3 M EDTA, 3 min of MW irradiation) possessed high SAR (49.2) on the surface (bulk SAR was 37.6) and well-preserved crystallinity (>95%). The optimal pH = 8–9 favored simultaneous desilication and chelation–dealumination. Catalytic testing of Z-HT-0.3-3 in the *ex situ* pyrolysis of high-density polyethylene (HDPE) allowed for a lower temperature (365 °C) for achieving the total substrate conversion compared to the parent ZSM-5 (410 °C). The application of Z-MW-0.3-3 resulted in a ~12% yield of oil, in which the aromatic compounds accounted for 78.7 area%.

In the catalytic co-pyrolysis of polymeric and biomass wastes, HDPE plays the role of a hydrogen source that causes an increase in the yield of MAHs, including BTX in comparison with pyrolysis of pure plant biomass. As demonstrated by Lin et al. [92], the *ex situ* catalytic co-pyrolysis of torrefied poplar wood sawdust (TPW) and HDPE (550 °C, 100 mL/min of N_2_, 3 g of the substrate (TPW: HDPE = 1), 3 g of catalyst, time 20min) in the presence of hierarchical HZSM-5 (synthesized by the common treatment with NaOH) allowed for an MAH yield of 71.75% at the torrefaction temperature of 260 °C.

The series of works by Li et al. [94,95,96] is worth mentioning since they proposed to create hierarchical porosity by the combination of microporous zeolite HZSM-5 and mesoporous silica MCM-41. MCM-41 does not exhibit Brønsted acidity and possesses low hydrothermal stability, while HZSM-5 has sufficiently high acidity and stability but suffers from coke deposition [97]. Thus, MCM-41 itself has low efficiency as a pyrolysis catalyst; its successful application requires modification with different metals (Al, Fe, Ni) [98,99,100,101,102]. However, in combination with microporous zeolite MCM-41, it can be beneficial for pyrolysis.

In ref. [94], HZSM-5 was treated with NaOH solutions of different concentrations (1.5, 2.0, 2.5 or 3.0 M at 40 °C for 60 min) to provide partial desilication and to create sites for attachment of MCM-41. MCM-41 was grown on the surface of HZSM-5 (source of silica) using CTAB with concentrations of 5, 10, 15 or 20wt.%. The proposed procedure [94] allowed for obtaining a hierarchical micro-mesoporous structure, whose pore volume increased with a corresponding increase in NaOH and CTAB concentrations. The resulting materials were utilized in the *ex situ* CFP of bamboo at 600 °C. It was shown that the highest yield of hydrocarbons (53.23%) was achieved while using HM-10%CTAB (HZSM-5 was treated with 2.0 M NaOH solution), which had the optimal ratio of HZSM-5 and MCM-41. The catalytic materials developed were also applied in the *ex situ* co-pyrolysis of waste greenhouse plastic films (W) (the hydrogen donor) and rice husk (R), a main source of oxygenates [95]. Under optimal temperature (600 °C), the hydrocarbon content reached 71.1% at the R/W mass ratio of 1:1.5. Later, Li et al. [96] reported the synthesis of a hierarchical catalyst using a treatment with TPAOH of HZSM-5 as a core and MCM-41 as a shell with variation in TPAOH and CTAB concentrations and synthesis conditions. The use of TPAOH instead of NaOH allowed one to preserve the structural characteristics of the catalyst. The samples of hierarchical catalysts were tested in the *in situ* microwave-assisted CFP (MACFP) of rice husk. The highest yield of hydrocarbons (60.5%) with the selectivity to MAHs equal to 43.5% was found at 2.0 M of TPAOH solution, 10 wt.% of CTAB, 24 h at 110 °C for digestion and crystallization. The optimal temperature of MACFP was 550 °C. For comparison, the yield of hydrocarbons in the presence of HZSM-5 under the same conditions was only 36.0%.

In contrast to the works of Li et al. [94,95,96], Yu et al. [103] synthesized MCM-41/ZSM-5 composites using tetraethoxysilane (TEOS) as a silica source, which allowed them to obtain MCM-41 microspheres with incorporated ZSM-5 particles (Figure 14). The series of catalysts with different ratios of MCM-41 and ZSM-5 were tested in the *in situ* CFP of miscanthus (600 °C, residence time 20 s, BCR 0.2). It was discovered that the BTX yield increased with the corresponding increase in the ZSM-5 content from 25% up to 100% (Figure 15). The total yield of hydrocarbons in the case of the composite with 75% of ZSM-5 was close to that of pure ZSM-5. It was proposed that the MCM-41 layer blocks the active sites on the outer surface of ZSM-5 responsible for deoxygenation reactions. In our opinion, this method results in the formation of a bulky layer of MCM-41 around the ZSM-5 particles. Thus, the relatively higher stability to coke deposition on the MCM-41-75%ZSM-5 surface as compared to pure ZSM-5 was the only advantage.

Table 1 summarizes the data of catalytic pyrolysis in the presence of hierarchical zeolites. As can be seen from Table 1, the most widespread approach for the synthesis of hierarchical catalysts for pyrolysis is chemical treatment, in particular, alkaline desilication.

It is worth mentioning a few other contributions to the field, which are outside of the above classification, particularly the catalytic treatment (upgrading) of pyrolysis bio-oils [104,105,106] or model compounds and their mixtures [107,108]. In the majority of relevant works [104,105,106,107,108], hierarchical ZSM-5 prepared via alkaline (NaOH) desilication was used. The overall advantage of hierarchical zeolites as compared to their parent materials is the increase in the aromatics yield, lower content of oxygenates in the upgraded bio-oil, and improved catalyst durability and recovery. Mesopores of hierarchical zeolites enhance the mass transfer of phenolic compounds, which can further generate aromatics.

#### 2.1.2. Metal-Doped Zeolites as the Pyrolysis Catalysts

Metal-doped natural aluminosilicates as well as synthetic zeolites are known to be the catalysts for pyrolysis [87,109,110,111,112,113,114,115,116,117]. In this review, we will only discuss hierarchical metal-doped zeolites because the microporous metal-containing systems are well described and analyzed.

There is a limited number of works where zeolites with hierarchical porosity and metal dopants are discussed [118,119,120,121,122,123,124]. Among the metal-containing hierarchical zeolites, the most popular systems are hierarchical ZSM-5 doped with Fe [120,123], Ni [121,122] and Ga [120,121]. Other non-noble metals, which can be impregnated into hierarchical zeolites, are Mg [118], La [119], Cu, and Zn [118,121].

Hernando et al. [118] reported the impregnation of hierarchical h-ZSM-5 and h-Beta with MgO (8.4–8.7 wt.%) or ZnO (9.7–10 wt.%). Hierarchical zeolites were synthesized via the bottom-up approach using TPAOH or TEAOH templates with the addition of a silanization agent, phenylaminopropyltrimethoxysilane (PHAPTMS). It is known that the metal deposition on zeolites changes the concentration and ratio of BAS and Lewis acid sites (LAS) on the surface [117]. The incorporation of MgO and ZnO caused a reduction in the BAS concentration and, at the same time, an increase in the LAS concentration, associated with Mg and Zn species. This resulted in a corresponding decrease in coke formation (2.3–2.7 wt.%) and higher catalyst stability, as compared to the metal-free zeolites during the *ex situ* pyrolysis of eucalyptus woodchips (500 °C, catalyst 1 g, BCR 5). The yield of bio-oil was nearly the same (about 30%) for both the parent zeolites and metal-containing samples; however, the percentage of oxygenates in the bio-oil composition decreased. On the other hand, the concentrations of aromatic hydrocarbons were negligible, especially for h-Beta-based catalysts, while oxygenated aromatics prevailed in the bio-oil composition.

Li et al. [119] carried out the *ex situ* (two-stage fixed-bed reactor) pyrolysis of rape straw at 500 °C using hierarchical HZSM-5 (Hi-ZSM-5) obtained via the treatment of a parent zeolite with Na_2_CO_3_ and impregnated with La (5 wt.%) (La/Hi-ZSM-5). In the case of La/Hi-ZSM-5, it was shown that the incorporation of La resulted in a noticeable increase in LAS and a slight increase in BAS concentrations. It was proposed that La ions interacted with the zeolite to form two stable structures of Z-La(OH)_2_^+^ and Z-LaO(OH), which possessed Lewis acidity. Thus, the increase in the LAS concentration resulted in an increase in the catalytic activity of La/Hi-ZSM-5, as compared to Hi-ZSM-5 (and also in comparison with the parent HZSM-5): the yield of gas-phase products increased from 44.81% up to 46.23%; the yield of the liquid phase decreased from 31.83% to 26.33%; the yield of the bio-oil organic phase decreased from 28.31% to 17.09%. It was proposed that secondary reactions, cracking of large molecules, can proceed in mesopores, while micropores are responsible for transformations of small molecules to olefins, aromatics, and non-condensable gases. At the same time, the yield of oxygenates and toxic coke decreased in the following range: HZSM-5 > Hi-ZSM-5 > La/Hi-ZSM-5.

Chen et al. [121] synthesized a series of metal (Ni, Ga, Cu, Zn)-containing catalysts based on the hierarchical ZSM-5 obtained by NaOH desilication. The *in situ* CFP of rice straw at 600 °C revealed that the effect of Me doping strongly depends on the metal loading. In particular, the impregnation with Ni and Cu at low loadings (0.1 wt.%) allowed for higher yields of aromatics (29.2% and 28.0%, respectively) in comparison with the metal-free hierarchical zeolite (27.4%). The incorporation of 0.1 wt.% of Zn and Ga had no effect. In the case of Ga, the optimal loading was 0.5 wt.%, which increased the yield of aromatics up to 28.1%. However, the higher metal loading resulted in a decrease in catalyst efficiency, likely due to the ability of chosen metals to replace H^+^ on the catalyst surface, reducing its acidity as well as its cracking ability. In a similar way, Dai et al. [120] revealed that impregnation with Ga vs. Fe (1 wt.%) in hierarchical ZSM-5 allowed for higher production of aromatics from pyrolysis of cellulose and lower coke formation.

In refs. [122,123], catalysts with high metal loadings were reported. Li et al. [123] obtained Fe(i)/Hie-ZSM-5 catalysts containing 2–8 wt.% of Fe. The highest selectivity (19.92%) to aromatic hydrocarbons was found for Fe(4)/Hie-ZSM-5 in catalytic pyrolysis of poplar wood [123]. Ni-loaded (2–11 wt.%) catalysts based on hierarchical ZSM-5 were also developed [122]. In the *in situ* CFP of torrefied corn cob at 550 °C, these catalysts allowed for an increase in the relative content of aromatics in the produced bio-oil from 46.42% up to 54.42% (for 8 wt.% Ni- hierarchical ZSM-5).

Noble metal catalysts are very rare in pyrolysis. For the last decade, we have found only one example, the work by Zhang et al. [124], who reported the synthesis of hierarchical ZSM-5 via the top-down approach (treatment with NaOH) with further impregnation with Ru (xRu-MZSM). Catalysts 2Ru-MZSM and 5Ru-MZSM were tested in the *in situ* pyrolysis of cellulose at 350–750 °C. The oxidation state of ruthenium changed during the pyrolysis process from RuO_2_ to the mixture of RuO_2_ and Ru^0^ in a ratio of about 3:1. The authors [124] proposed that the *in situ* generated Ru^0^ was responsible for H-transfer, while strong LAS derived from RuO_x_ enhanced the dehydrogenation process (Figure 16). Thus, a higher total yield of aromatics (16.8%) can be achieved in comparison with the parent metal-free ZSM (3.6%) at 650 °C.

There is an example of hierarchical HZSM-5 impregnated with Sn, Cu, Ni, and Mg (1 wt.%) employed in bio-oil upgrading [125]. Though metal-doped zeolites resulted in a lower yield of aromatics in comparison with the pure Meso-HZ40, the main improvement was the decrease in the percentage of oxygenates. Among the Meso-MeHZ40 samples synthesized, the higher deoxygenation rate was found using the Mg-loaded hierarchical zeolite. This is because Mg cations act as additional LAS, accelerating the deoxygenation reactions. At the same time, Mg-doped zeolites contained the lowest amount of BAS, which resulted in a decrease in coke formation from PAHs. It was also mentioned [125] that the highest deoxygenation rate could be related to the promotion of esterification reactions in the case of Meso-MgHZ40.

Table 2 summarizes the data on catalytic pyrolysis in the presence of metal-doped hierarchical zeolites.

Table 2 demonstrates that Ga, Fe, and Ni are the most widely used metal dopants, which are typically deposited on the desilicated hierarchical ZSM-5 with the MFI framework type. In general, the introduction of metals allows for the following: (i) the reduction in the amount of toxic coke and PAHs due to the decrease in the BAS concentration, which, in turn, results in an increase in the catalyst lifetime; (ii) lower concentration of oxygenates in the bio-oil composition; (iii) the increase in activity due to the higher concentration of LAS; (iv) in most cases, the increase in the yield of light aromatics in comparison with the initial metal-free hierarchical zeolites.

### 2.2. Transformation of Alcohols to Fuel Hydrocarbons, Aromatics and Olefins

#### 2.2.1. Alcohols to Fuel Hydrocarbons and Aromatics

The catalytic conversion of alcohols (for example, MeOH) to hydrocarbons is another example of industrially important processes, which can benefit from the utilization of zeolite with hierarchical porosity. The methanol-to-hydrocarbon (MTH) process has been known for several decades. Depending on the preferential products of the MeOH conversion, the following processes can be highlighted: methanol-to-aromatics (MTA), methanol-to-olefins (MTO), and methanol-to-gasoline (MTG). The mechanism of MeOH conversion on the zeolite surface is rather complex and requires Brønsted acidity. A general scheme of the MTH process suggested for the ZSM-5 involves a dual-cycle reaction network (Figure 17) [126]. As emphasized by Tian et al. [127], the concentration of BAS plays different roles in the alkenes-based cycle and aromatics-based cycle: high density of BAS can accelerate the formation and accumulation of cyclic compounds, while in the presence of a low-acidic catalyst, the formation of olefins mainly follows methylation and a cracking route [127].

Moreover, Yarulina et al. [128] demonstrated that the pathway of the MeOH transformation and the set of obtained products depend on the zeolite topology (Figure 18). In 1D 10-membered ring structures, low-molecular-weight alkene intermediates dominate with a negligible production of aromatics. The increase in pore/cavity sizes allows for the production of an aromatic-rich mixture, unless there are no limitations for their diffusion into the gas phase [126]. A further increase in pore/cavity sizes leads to the formation of condensed aromatics, which serve as coke precursors [129].

Regarding the coke formation in the MTH process, two main routes were proposed [127]: (i) transformation of the hydrocarbon pool (HCP) intermediates (HCP is the adsorbed or trapped hydrocarbon species) [130]) and (ii) conversion of olefin products without involving HCP intermediates. HCP acts as a co-catalyst through ongoing methylation and dealkylation reactions, releasing mainly ethene and propene. The first route can be restricted by adjusting the sizes of zeolite cavities.

Thus, the MTH process can be complicated by the variety of reaction modes (influence of diffusion limitations), as well as the dynamic evolution of the catalytic surface [131]. Regarding the diffusion limitations and the related deactivation limitations, the development of hierarchical zeolites can have a positive impact on the catalyst selectivity and lifetime [132,133]. Zhou et al. [132] found the relationship between the degree of utilization (η) of zeolite and the Thiele modulus (φ); the latter allows one to quantify the degree of diffusion limitations.

For the parent zeolite (ZSM5-P), the lowest degree of utilization (32.1%) was found in the case of the isomerization reaction. In contrast, in the case of MTH, the utilization degree reached 94.5%, indicating fast mass transfer in pores. When the effective diffusion length was increased (the crystal size of the zeolite was enlarged from 120 nm up to 250 nm), the degree of utilization of the MTH reaction decreased to 82% (Figure 19) [132].

Thus, for the large zeolite crystals, the formation of hierarchical porosity is feasible, since it can promote mass transfer and increase the selectivity to MTH products. It was shown [132] that the utilization degrees for methylation, isomerization, and MTH followed the same order: bulky crystals (ZSM5-250 nm) < small crystals (ZSM5-P) < hierarchical small crystals (ZSM5-H). It was also stated [133] that the hierarchical zeolites delayed the onset of kinetically relevant deactivation, despite the accumulation of more coke than in microporous zeolites [133].

In the last decade, there has been a limited number of works devoted to the use of hierarchical zeolites in the reaction of alcohols to hydrocarbons [134,135,136,137,138,139,140]. That is likely due to the relatively low diffusion limitations for the MTH process (see the above discussion). In the majority of papers [135,136,137,138,139,140], ZSM-5 with the MFI framework structure was used. The composite intergrown materials ZSM-5/ZSM-11 were also reported [134,141]. ZSM-11 belongs to the family of MEL framework-type zeolites that is close to the MFI family but has a different channel structure: ZSM-5 consists of intersectional straight and sinusoidal 10-MR channels, while ZSM-11 is formed by crossing straight 10-MR channels (Figure 20) [142,143]. Moreover, ZSM-5 and ZSM-11 differ in the location of Al atoms. In HZSM-5, Al atoms preferably occupy the intersectional T sites; in HZSM-11, Al atoms are mainly located in the straight 10-MR channels. For the MTO reaction, it was proposed [143] that the acid sites located in the intersections lead to an increase in the aromatics yield, while the straight channels of ZSM-11 promoted the alkene cycle [143]. At the same time, channels of ZSM-11 can favor the easier diffusion of C_7_ and C_8_ aromatics out of the ZSM-11 in comparison with ZSM-5 [142].

The hierarchical composite ZSM-11/5 zeolite was synthesized by Wang et al. [134] directly via the addition of CTAB in a seed-induced method in the absence of templates. In the MTH process, the addition of CTAB caused an increase in the C_5+_yield due to the evolving stronger acid sites, which are responsible for the conversion of light intermediates into higher hydrocarbons (≥C_5_). The stability of the hierarchical ZSM-11/5 composite was also higher than that of commercial ZSM-5 with a similar SAR [134].

The series of recent works of Anekwe et al. [137,138,139,140] reported the application of metal-doped ZSM-5 in the conversion of different alcohols (MeOH, ethanol (EtOH), 1-propanol (PrOH)) to hydrocarbons (MTH, ETH, and PTH processes, respectively). ZSM-5 was synthesized in one stage (without post-treatment) via the hydrothermal method with the addition of tetrapropylammonium bromide as a structure-directing agent. The surface morphology of the ZSM-5 obtained was similar to that reported by Ding et al. [35], allowing one to consider hierarchical porosity. Different metals (Co, Fe, Ni) were deposited on ZSM-5 by the impregnation method [137,138,139,140]. In the alcohols-to-hydrocarbons processes, the highest selectivity (30–70%) was found for C_5_–C_8_ hydrocarbons independently of the presence and nature of a metal dopant. In the MTH, ETH, and PTH processes, different selectivity to BTX and high hydrocarbons (C_9–12_ and C_12+_) was observed. In the MTH and PTH reactions, for example, most catalysts did not allow for a higher BTX yield, as compared to pure zeolite. The only exceptions were Co/HZSM-5 in the ETH and Ni/HZSM-5 in PTH processes, for which the BTX yield was noticeably higher, as compared to other catalysts. The coke deposition behavior was also different from the catalyst to catalyst and from the substrate to substrate. In general, it was concluded that the increased metal doping accelerated the coke yield due to the increase in the concentration of acid sites that can result in faster catalyst deactivation. In our opinion, the only advantages of metal doping here can be shifts in the product distribution and better regeneration due to the easier combustion of coke, as compared with metal-free zeolite.

Unlike the MTH process, noticeable improvements in MTA can be achieved through hierarchization and metal doping of the MFI zeolites. The introduction of metals enhances the dehydrogenation of intermediate alkenes/alkanes, with a release of H_2_, that is crucial for the MTA process. In addition, synergy between BAS and Lewis metal sites is important to achieve high aromatics selectivity [144].

Ma et al. [145] emphasized that the reactions of olefin oligomerization and cyclization can be enhanced via the increase in the Al_pairs_(Al-O-(Si-O)_1,2_-Al) content, which can likely stabilize the transition states of aromatization, and, hence, more BTX can be obtained. Using recrystallization and post-realuminization methods, hollow-structured ZSM-5 zeolites with more Al_pairs_ in the framework and more Al in the intersection cavities were synthesized. It was shown that the catalytic properties of the materials obtained can be further increased by the introduction of Ga^3+^ species via ion exchange. In this way, 95% of the propane conversion at 70% of aromatics selectivity was achieved at 540 °C and 1 atm [145]. In a similar way, hierarchical Ga-promoted zeolites were successfully applied in the MTH process. Liu et al. [146] synthesized three zeolite samples: (i) nanosized, spherical, and purely microporous ZSM-5; (ii) hexagonal, prismatic and hierarchical ZSM-5 (glucose was used as an additive during the zeolite synthesis to create mesoporosity); and (iii) coffin-shaped and hierarchical ZSM-5. The introduction of mesoporosity was found to reduce the diffusion barrier for the products and catalyst lifetime. However, an increase in crystal length (prolongation of the diffusion path of olefins) higher than the certain value constrained the aromatics yield: for ZSM-5 (m-ns) (110 nm) and ZSM-5 (h-hexag) (450 nm), aromatics selectivity was 15% and 20%, respectively, while for the ZSM-5 (h-coffin) (4600 nm), aromatics selectivity was only 22%. Ga-doped samples surpassed the corresponding metal-free ZSM-5 samples in terms of activity and selectivity. The highest selectivity to aromatics (∼34%, 73% of which belonged to BTX) and the lowest propylene/ethylene ratio of 0.8 were observed for Ga/ZSM-5 (h-coffin) zeolite (Figure 21).

Liu et al. [146] proposed that the introduction of Ga species could neutralize, to some extent, the strong BAS of zeolite, resulting in additional medium acidity. This, in combination with the hierarchization and changes in crystal sizes, might cause an increase in formaldehyde production [147], reduction in hydrogen transfer and cracking of C_5+_ hydrocarbons, as well as an enhancement in oligomerization and cyclization/aromatization of short-chain olefins (Figure 21). It was also concluded that crystal size plays a more significant role than the hierarchization degree in yielding aromatics.

Zinc is another metal that has a positive impact on the catalytic behavior of hierarchical zeolites [141,142,143,144,145,146,147,148]. In the series of works conducted under the guidance of Wei F. [148,149,150], the authors reported the synthesis of nanosized ZSM-5 crystals organized in hierarchical aggregates [148,149] or in the ordered micro-meso-macroporous structure consisting of highly interconnected MFI zeolite nanorods (Figure 22) [150].

The hierarchical zeolites developed were impregnated with zinc compounds (2 wt.% of Zn) and tested in the MTA process (475 °C; ambient pressure; weight hourly space velocity (WHSV) = 0.75 h^−1^). Zn-containing catalysts based on the nanosized ZSM-5 revealed remarkable MeOH conversion (>96%) with a total aromatics selectivity of >62% for the 42 h time on stream [148,149]. The catalyst stability depended on the zeolite SAR. The slowest deactivation (catalyst lifetime >75 h) was found in the case of SAR = 60. For comparison, for common zeolite ZSM-5, the catalyst lifetime was only 3 h [148]. Zn supported on the zeolite nanorod arrays [150] also showed high MeOH conversion (>95%) with a selectivity to aromatics of >60% for the initial 50 h time on stream. However, this micro-meso-macroporous catalyst allowed for much higher capacity for coke accumulation in comparison with Zn/MFI nanoparticles that resulted in about five-fold higher stability.

In the work of Shen et al. [151], hierarchical ZSM-5 samples were obtained via the top-down approach, combining the treatment with NH_4_F and NaOH, which allowed for a trimodal pore structure composed of micropores (0.55 nm and 0.85 nm) and mesopores (6.5 nm). The formation of hierarchical porosity, along with the loading of H-ZSM-5 with ZnO (0.2 wt.%) by the ball-milling method, resulted in an increase in the catalyst lifetime up to ∼140 h and the BTX yield > 40% for about 70 h that was several times higher than that for metal-free H-ZSM-5 (0.1 MPa, 733 K, 80 kPa of MeOH, flow rate 25 mL/min). The authors [151] stated that the MTH reaction route depended on the sizes of ZnO particles. In the case of small subnanometer ZnO species, dehydrogenative aromatization became a major route, since the active ZnO sites were responsible for the recombinative desorption of H atoms as gaseous H_2_.

It is noteworthy that not only C_1_–C_3_ alcohols but also higher-molecular-weight alcohols can enter the aromatization reaction. An interesting work was published by Wang et al. [152], who converted bio-based crude glycerol to aromatics (450 °C, WHSV = 0.71 h^−1^) in the presence of hierarchical HZSM-5 (1.2 g) synthesized by the seed-assisted hydrothermal approach using CTAB and [3-(trimethoxysilyl)propyl]dimethyloctadecylammonium chloride (TPOAC) as soft mesopore structure-directing agents. TPOAC (template) allowed for the maximum yield of BTX (38.5%) and the longest lifetime (23 h) due to the facilitated mass transfer into the interconnected mesopores [152]. We believe that metal doping would allow for further improvements in the catalytic properties of hierarchical HZSM-5 in the glycerol-to-aromatics process.

#### 2.2.2. Methanol to Olefins

Light olefins, e.g., propylene, can be synthesized by three routes: (i) propane dehydrogenation [153,154,155], (ii) the Fischer–Tropsch olefin process [156,157,158], and (iii) the MTO process. The MTO process is one of the most important industrial reactions to produce basic petrochemicals from non-oil feedstock, which typically proceeds in the presence of the commercial SAPO-34 (silicoaluminophosphate) catalyst [127,159,160]. The general issue of the MTO process is coke deposition, causing catalyst deactivation, which can be partially offset by the decrease in Si content [160] and by the hierarchization of SAPO-34 [161]. In addition, surface poisoning can be applied to suppress the undesired hydrogen transfer reactions [161]. Hierarchization of SAPO-34 by the top-down (treatment with citric acid) [162] or bottom-up (hydrothermal treatment with the addition of secondary silica) [163] methods was reported. The treatment of SAPO-34 with citric acid allowed nearly double external porosity, which, in turn, resulted in an increase in catalyst stability in the MTO reaction (450 °C, WHSV = 1.5 h^−1^) by ~60%, as compared to the parent microporous material. This can be attributed to the expanded space for coke deposition and the five-fold reduction in the diffusion resistance [162]. Hierarchical SAPO-34 obtained by the bottom-up method [163] also allowed for a decrease in diffusion limitations and an increase in olefin selectivity by about 1.5% (MTO conditions: 733 K, WHSV = 6.0 h^−1^) [163]. Using the bottom-up approach, Zhu et al. [164] synthesized the hierarchical zeolite SSZ-13 that was the isostructural phosphorous-free aluminosilicate analogue of SAPO-34. Due to the higher acidity of this zeolite (SSZ-13) compared to SAPO-34, the MTO reaction could proceed at much lower temperatures (350–375 °C) than in the case of SAPO-34 [164].

In addition to SAPO, conventional zeolites, such as ZSM-5, can serve as the MTO catalysts [130,165] and can be subjected to hierarchization. For example, Li et al. [166] developed the hierarchical micro-meso-macroporous ZSM-5 material while using short-chain organosilane (3-aminopropyltrimethoxy-silane (APTES)), which was anchored on the surface of nanocrystals and prevented their excessive growth. The APTES addition resulted in the formation of the intercrystal mesoporous and macroporous structure, which provided high efficiency in the MTO reaction (470 °C, 1 bar, MeOH: H_2_O = 1:1, WHSV = 8 h^−1^). The catalyst lifetime exceeded 89 h (at the MeOH conversion > 95%), i.e., eight-fold increase compared to commercial ZSM-5 [166].

### 2.3. Cracking, Hydrocracking, and Hydroisomerization

#### 2.3.1. Catalytic Cracking and Hydrocracking

Cracking and hydrocracking are widespread reactions, which can be part of such complex processes as pyrolysis, MTO, and MTH. They can also be considered as separate reactions, e.g., for conversion of waste plastic (i.e., HDPE, low-density polyethylene (LDPE), and PP) to hydrocarbons [167,168,169,170,171,172,173,174]; transformation of alkanes to light olefins [175,176,177,178,179,180]; conversion of substituted or condensed aromatics to light aromatics (i.e., BTX) [181,182,183,184]; processing of vegetable oils to hydrocarbons [185,186].

In the case of cracking, BAS are the main active sites stabilizing the carbonium ion and carbenium ion transition states (TSs) (Figure 23). LAS may influence reaction steps, enhance absorption, and influence directly or indirectly the properties of BAS. During the cracking process, numerous side reactions (isomerization, hydride transfer, dimerization, cyclization, aromatization, and oligomerization) also take place [187]. The first step of alkane cracking, the formation of a carbenium intermediate from a saturated compound, is under discussion. Presumably, LAS can abstract a hydrogen atom from an alkane molecule. The resulting alkene can be further transformed via the carbenium mechanism on BAS [188]. Alternatively, carbenium intermediates can be produced from alkene impurities in the alkane feed or by non-catalytic thermal cracking [189].

Similarly to the MTH process, the product selectivity in the case of cracking is determined by the balance between two cycles (see Figure 17): an alkene cycle, which is dominated by zeolite channels, and an arene cycle, dominated by the intersections [189]. Particularly, propylene is a preferential product of the alkene cycle rather than ethylene, although both olefins could potentially stem from the arene cycle [189].

The size and shape of zeolite pores along with the strength and location of BAS play an important role in the cracking process. Pore sizes influence not only the rate of C-C bond cleavage but also the side and sequential reactions [187]. Thus, in the cracking reactions, hierarchical zeolites can benefit as compared to the traditional microporous zeolites, for the same reasons as for other processes discussed above: the intensification of mass transfer, especially in the case of bulk molecules, and an increase in selectivity and catalyst lifetime [190]. Below, we will discuss some examples of the utilization of hierarchical zeolites in the cracking and hydrocracking processes.

Different hierarchical zeolites (ZSM-5, Y, β) can be employed in the cracking and hydrocracking of polymeric waste. Tarach et al. [167,168] investigated the catalytic cracking of HDPE and LDPE in the presence of hierarchical ZSM-5 obtained by the top-down approach, desilication with either NaOH or NaOH/TBAOH [167] and a combination of desilication (NaOH) with mild dealumination (HNO_3_) [168]. Cracking of HDPE and LDPE was investigated using a TGA method at the polymer/zeolite ratio of 3:1. It was shown that the treatment of parent microporous ZSM-5 with NaOH/TBAOH allows for a double increase in the accessibility of BAS in comparison with the sample treated with NaOH alone. The higher acidity and better accessibility of acid sites likely caused the formation of more active alkylcarbenium or alkylcarbonium ions on the catalyst external surface, which were pivotal at the early stage of HDPE cracking. Cracking of HPDE was more influenced by zeolite hierarchization than for LDPE, since HDPE is a linear polymer in contrast to highly branched LDPE [167]. A combination of desilication and dealumination resulted in higher accessibility of acid sites due to unblocking of pores by the strong Lewis residues, which, in turn, allowed for a decrease in temperature necessary for reaching 50% conversion from 325 °C to 300 °C [168].

Depending on the catalyst composition, hydrocracking can proceed via distinct mechanisms. In the presence of metal-free zeolites, the reaction takes place via the carbocation mechanism. When metal-doped zeolites are used, the metal favors dehydrogenation and hydrogenation reactions, while the acidic centers are responsible for cracking and isomerization reactions [169]. Hydrocracking is a promising process, more advantageous than pyrolysis, for the transformation of polymeric waste into value-added hydrocarbons, since it occurs at lower temperatures and allows one to produce liquid fuels without the necessity of further upgrading. In addition, the presence of hydrogen decreases the coke deposition, giving extra lifetime to the catalyst [170].

Ni and Pt are the most popular metals for hydrocracking of polyolefins [170,171,172]. Recently, it was shown [174] that the nanoscale proximity of metal and acid sites is pivotal for the successful hydrocracking of macro-molecular polyolefins to gasoline. Among the hierarchical zeolites, the samples obtained by desilication methods reveal the best properties. Desilicated HZSM-5 loaded with 5 wt.% of Ni provided 100% conversion of HDPE to gaseous products (C_1_–C_5_) for 60 min at 260 °C and 20 bar of H_2_ while suppressing coke formation [170]. Moreover, an optimal balance between metal sites and acid sites on Ni-impregnated zeolites led to higher selectivity to aromatics while decreasing naphthene selectivity and a higher *iso/n*-paraffins ratio in the gasoline-range products obtained from HDPE hydrocracking [171,172]. The importance of the proper metal–acid balance was discussed in ref. [173]. The authors synthesized a cerium-promoted Pt/HY catalyst with hierarchical porosity for hydrocracking of different polyolefins. The moderate acidity of the catalyst developed allowed one to avoid the over-cracking of the intermediate hydrocarbons to light C_1–4_ products. The introduction of Ce into the catalyst provided oxygen vacancies for Pt adsorption, leading to a significantly improved Pt dispersion. Thus, using the optimal Pt-3Ce/HY catalyst (0.5 wt.% of Pt, 3 wt.% of Ce), a high yield of C_5–12_ hydrocarbons (up to 85 wt.%) was achieved from LDPE for 2 h at 280 °C and 2 MPa of H_2_ [173].

In the case of catalytic cracking of hydrocarbons (*n*-hexane, *n*-heptane, *n*-decane, *n*-dodecane, etc.) to light olefins, the catalysts based on ZSM-5, including the hierarchical samples obtained by the one-step synthesis with various templates [175,176,177,178] as well as composite zeolites ZSM-5/MCM-41 [179], are the most prevalent. The main finding of these works was the direct correlation between the hierarchization degree and the observed catalytic activity and zeolite lifetime. Hierarchical pores could also inhibit the occurrence of the hydrogen transfer reaction and improve selectivity to light olefins. Moreover, a monomolecular pathway of cracking (Figure 24) prevails in the case of hierarchical zeolites, since the residence time of primary olefin products is likely reduced [178]. The major drawback of mesoporous zeolites in the cracking reactions is their relatively low thermal stability. This shortcoming can be overcome by so-called “pillaring” [177] (see the Introduction), which stabilizes mesopores, intensifies the product diffusion, and restricts secondary reactions and coke formation.

Parsapur et al. [180] recently reported the application of hierarchical FAU-type zeolites for hydrocracking of vacuum gas oil (VGO). A non-classical top-down approach was developed for the synthesis of hierarchically ordered FAU-type frameworks. It included the formation of mesoporosity by the structural reorganization of the parent zeolite nanocrystals (Figure 25) in the presence of surfactants and inorganic salts. The latter stabilized the mesophase transitions via ion-specific interactions.

The catalysts were prepared by mixing zeolites (30 wt.%) with alumina (70 wt.%), followed by the impregnation with Ni and Mo species and sulfidation. The NiMoS/ZAK-1 sample, possessing a high BAS/LAS ratio and 3D mesostructure, demonstrated the VGO conversion of 91.5% at 400 °C and 90 bar (for the commercial catalyst, the VGO conversion reached only 71.2%). In addition, lower internal coke deposition (15–20%) compared to the parent zeolite (35%) was achieved due to the enhanced diffusion properties [180].

Among the catalysts for the cracking of aromatics to BTX [181,182,183,184], we would like to emphasize hierarchical Beta zeolites [183,184]. Because Beta zeolite has lower stability to desilication or dealumination treatments as compared to other zeolites, such as ZSM-5 or Mordenite [183], the secondary porosity should be created using a template method rather than chemical etching. Recently, Hu et al. [184] reported a method for obtaining three-dimensionally ordered macroporous Beta (3DOM-Beta) zeolite based on ordered poly(methyl methacrylate) (PMMA) microspheres as a template (Figure 26).

3DOM-Beta was used as a support for the NiW sulfide catalyst intended for the reaction of hydrocracking of 1-methylnaphthalene at 420 °C and 6.0 MPa. NiW/3DOM-Beta contained the highest amount of BAS among the catalysts studied, thus promoting the ring opening and side-chain breaking reactions, resulting in higher selectivity (40.7%) and yield (40.2%) of BTX. A high mesopore volume (0.25 cm^3^/g) and excellent pore connectivity greatly shortened the diffusion reaction path and inhibited the condensation of PAHs, resulting in the lowest amount of coke (6.7 wt.%) [184].

In conclusion, it should be underlined that during the cracking process, the coke formation is mainly associated with the accumulation of alkylated aromatics [191]. This, in turn, depends on the zeolite acidity and porosity. However, in most cases, the mesopore generation alone does not efficiently reduce the total amount of coke (it mainly changes its location) [188,192]. The formation of coke can be effectively suppressed by decreasing the concentration of internal defect sites (i.e., internal silanols) (Figure 27), via zeolite annealing [192].

#### 2.3.2. Catalytic Hydroisomerization

Hydroisomerizaion is an industrially important process intended to increase the octane number and the improvement in the quality of gasoline. This reaction can also produce second-generation green biodiesel from the long-chain alkanes, products of the hydrodeoxygenation of vegetable oils. Hydroisomerizaion often accompanies hydrocracking; hence, they can be combined under the general term “hydroconversion” [193]. Thus, hydroisomerization follows a mechanism similar to hydrocracking (see Figure 23), involving BAS-stabilized carbenium intermediates. However, the main goal of the hydroisomerization process is to minimize the yield of cracking products, i.e., light hydrocarbons C_1_–C_5_. To achieve high selectivity to *iso*-products and to suppress the secondary cracking, the catalyst acidity along with the diffusion properties should be carefully regulated. Moreover, fast hydrogenation and dehydrogenation steps should be provided. Hierarchical metal-doped (Pd, Pt, Ru) zeolites, such as ZSM-12 [194], SAPO [195,196], Y [197], Mordenite [198], ZSM-5 [199,200], and ZSM-5/MCM-41 composites [201], are widely used in hydroisomerization [196,197,198,199,200,201]. The important factor here is the presence of noble metals, which can activate H-H, C-C, and C-H bonds, preventing the deposition of heavy hydrocarbons and coke formation and, hence, increasing the catalyst lifetime [197].

For tuning the support acidity, zeolites can be additionally modified with different non-noble metals. For example, the mesoporous Pt/SZ-mHY catalyst containing zirconia sulfate allowed for ~73 mol.% conversion of *n*-heptane and about 58 mol.% of *n*-hexane (160 °C, 15 bar, liquid hourly space velocity (LHSV) of 1 h^−1^), allowing for a multi-fold increase, as compared to the microporous catalyst sample [197]. Al-Rawi et al. [198] studied Ba- and Sr-modified zeolites as supports for bimetallic (Pt-Zr, Pt-Ru) and trimetallic (Pt-Ru-Zr) catalysts of *n*-hexane hydroisomerization. The highest yield of isomerization products (~41%) was found for Pt-Zr supported on Sr-ZSM-5, while Ru-containing catalysts produced significant amounts of cracking products (C_1_–C_5_).

For the hydroisomerization of long-chain hydrocarbons, in contrast to the C_6–7_alkanes, the enhanced Brønsted acidity and hierarchical porosity with the shortened channel length are pivotal [193,194]. BAS are responsible for the hydroisomerization of *n*-alkanes, which involves the isomerization and cracking of olefin intermediates [195]. At the same time, milder acidity can effectively inhibit side cracking reactions [195,196], which makes such supports as hierarchical SAPO preferable in comparison with traditional highly acidic zeolites.

### 2.4. Hydrodeoxygenation

Hydrodeoxygenation is another reaction belonging to the group of processes intended for fuel production, which can benefit from the utilization of hierarchical zeolites. One of the examples of such reactions is the hydrodeoxygenation of biomass-derived levulinic acid (LA) to gamma-valerolactone (GVL) and further to valeric fuels [200,201]. The recent work of Li et al. [202] demonstrated that hierarchical HZSM-5 loaded with Ni is a promising catalyst for the tandem reaction of LA conversion to the ethyl valerate (EV), an environmentally friendly oxygenated additive for the liquid fuels. This tandem reaction occurs via the synergetic catalysis of C=C bond hydrogenation and C-O bond cleavage. In contrast to microporous zeolite, hierarchical HZSM-5 provided a catalytic lifetime of 180 h, which was noticeably higher than that of the microporous counterpart (20 h). This outstanding result was achieved through the combination of secondary porosity and mild acidity, which reduced coke formation.

Other biomass-derived molecules, such as fatty acids [203,204] and lignin derivatives [205], can also undergo hydrodeoxygenation in the presence of hierarchical metal-doped materials. In these reactions, improved catalytic selectivity and stability are achieved via a combination of hierarchical porosity with mild acidity. Ding et al. [204] emphasized the importance of the spatial segregation of different sites for the realization of cascade processes. For example, Pd nanoparticles located in the mesopores of hierarchical ZSM-5 should be separated from acid sites belonging solely to micropores, facilitating hydrodeoxygenation at the simultaneous suppression of undesirable decarboxylation and decarbonylation [204]. For the hierarchical Ni/GaMFI catalyst, the regulation of the distribution of Ni nanoparticles and acid sites allowed high activity (conversion > 99%) and selectivity (>99%) in the reaction of the vanillin conversion to 2-methoxy-4-methylphenol [205]. The introduction of Ga resulted in an increase in the amount of acid sites for the simultaneous regulation of the distribution of Ni species and the metal oxidation state (higher share of Ni^0^ was obtained), which favored the hydrodeoxygenation of vanillin.

## 3. Conclusions

The hierarchization of zeolites is an important tool for the controlled synthesis of heterogeneous acid catalysts. Hierarchical porosity can be reached via numerous methods, which combine chemical etching with the addition of surfactants, pore-directing agents, and stoppers, as well as various soft and hard templates. Despite this diversity of approaches, to date, the most popular method is the alkaline desilication of zeolites, in particular, ZSM-5.

In this review, we discussed catalytic processes, which can benefit from the use of hierarchical zeolites, especially those with metal doping. These processes include pyrolysis, conversion of alcohols to hydrocarbons, aromatics and olefins, cracking and hydrocracking, hydroisomerization, and hydrodeoxygenation. For each reaction, hierarchical zeolites show advantages in comparison with parent microporous materials. At the same time, there are some common features as well as differences in the preferable types of zeolite-based catalysts for the above reactions.

In the case of pyrolysis, hierarchical ZSM-5 is a popular catalyst, for which the balance of surface acidity and the share of mesoporosity are of high importance. The excessive Brønsted acidity and mesoporosity result in unfavorable deposition of PAHs and coke, since mesopores, in general, favor secondary cracking. In contrast, medium mesoporosity and mild acidity with the high concentration of LAS allow for an increase in the substrate conversion and the yield of MAHs. Moreover, the existence of mesopores facilitates the access of the reagents to the internal microporous regions of zeolite species and provides faster diffusion of pyrolysis products. Thus, the main advantage of hierarchical zeolites is their higher durability due to the reduced deposition of external and toxic coke. The decrease in coke formation can be achieved by the formation of composite materials ZSM-5/MCM-41, since MCM-41 does not contain BAS. However, a certain ratio between microporous ZSM-5 and mesoporous MCM-41 is required to achieve the maximum yield of desired products, e.g., BTX.

The addition of metals (e.g., Ga, Fe, Ni, etc.) allows for a further improvement in pyrolysis catalysts based on hierarchical zeolites in terms of the following: (i) the reduction in the amount of toxic coke and PAHs due to the decrease in the BAS concentration; (ii) lowering the concentration of oxygenates in the bio-oil composition; (iii) the increase in activity due to the higher concentration of LAS; (iv) the increase in the yield of MAHs.

For other processes, such as MTH, mesoporosity and metal doping are not as important as for pyrolysis in terms of diffusion limitations and the BTX yield. A benefit is the increase in the catalyst lifetime. However, in the case of the MTA reaction, a noticeable increase in the yield of BTX can be achieved, especially when applying metal-doped (Ga, Zn) hierarchical materials with medium acidity and relatively long channels, which increase the pathway for olefin intermediates. For the MTH and MTA processes, ZSM-5 is also the most widespread zeolite. In contrast, for the MTO reaction, the hierarchical SAPO having milder acidity as compared to ZSM-5 is the best choice.

The catalysts intended for hydrocracking and hydroisomerization contain mainly Ni or noble metals (Pt, Pd, etc.). They suffer less from coke deposition, since these processes proceed under the hydrogen atmosphere. Thus, hierarchical metal-doped zeolites with higher acidity can be utilized. In this case, the effect of hierarchization results in better diffusion properties of such zeolites. In general, hierarchical porosity noticeably increases the catalyst lifetime.

## 4. Challenges and Outlook

The development of catalysts based on hierarchical zeolites is a growing field, impacting various processes of both academic and industrial importance. Despite significant accomplishments in this field in the last decade, the major challenges include the maintenance of structural stability of hierarchical zeolites, moderate acidity and its retention, control over the rate of coke formation and its location, and the increase in the activity of the catalysts based on hierarchical zeolites.

To address the above challenges, the following actions can be considered:(i)Development of hierarchical structures from 2D layers, which possess better hydrothermal stability;(ii)Combination of zeolites with different silica-based materials in composite hierarchical structures with controllable acidity;(iii)Modification of zeolite surface with different silica or phosphorous species and control over the defect concentration during the synthesis of the hierarchical catalytic material.

Despite many accomplishments in the development of sophisticated zeolites, in the majority of catalytic processes, the simplest hierarchical systems are still used, e.g., those obtained by alkali treatment of parent zeolite materials. We believe that the main obstacle for the wide implementation of hierarchical zeolites with uniform and well-defined porosity is the complexity and high cost of their synthesis. Thus, the use of new templates and/or a preferable shift to the template-free methods of zeolite syntheses as well as the development of one-stage methods of hierarchical zeolite syntheses are the most promising avenues.

Zeolites with hierarchical porosity, including the metal-doped ones, can be successfully used in catalytic reactions for the production and upgrading of liquid fuels, namely pyrolysis of biomass and polymer waste; conversion of alcohols into fuel hydrocarbons, aromatic compounds and olefins; cracking and hydrocracking of polymer waste and hydrocarbons; and hydroisomerization.

## Figures and Tables

**Figure 1 molecules-30-03798-f001:**
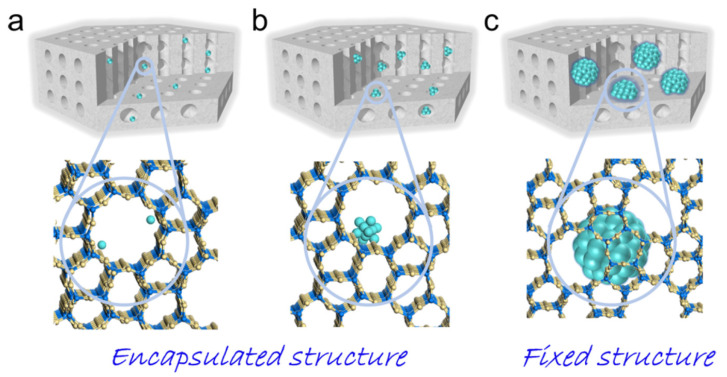
Metal@zeolite with encapsulated and fixed structures. (**a**) Isolated metal sites and (**b**) metal nanoclusters encapsulated in the micropores. (**c**) Metal nanoparticles fixed in the zeolite crystals [9].

**Figure 2 molecules-30-03798-f002:**
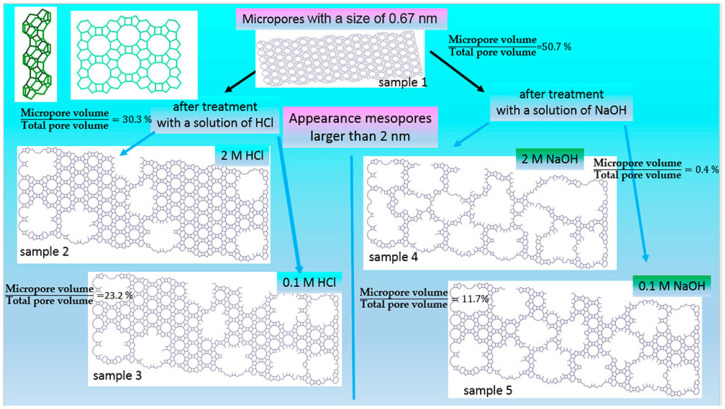
The model structures of samples. Samples 2, 3 were treated with solutions of HCl at different concentrations to dealuminate them. Samples 4, 5 were treated with solutions of NaOH at different concentrations to recrystallize and desilicate them [18].

**Figure 3 molecules-30-03798-f003:**
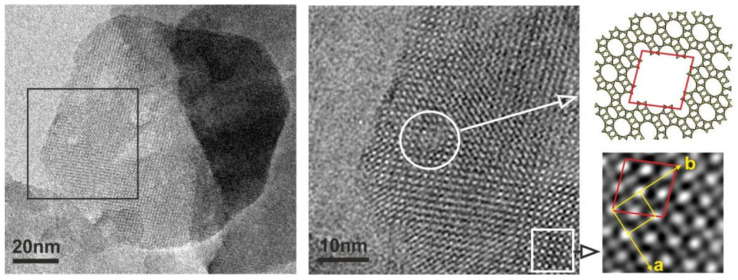
A high-resolution TEM image of MOR-A-AF demonstrating the presence of a defect-induced mesopore. Reproduced with permission from [20], published by RSC, 2020.

**Figure 4 molecules-30-03798-f004:**
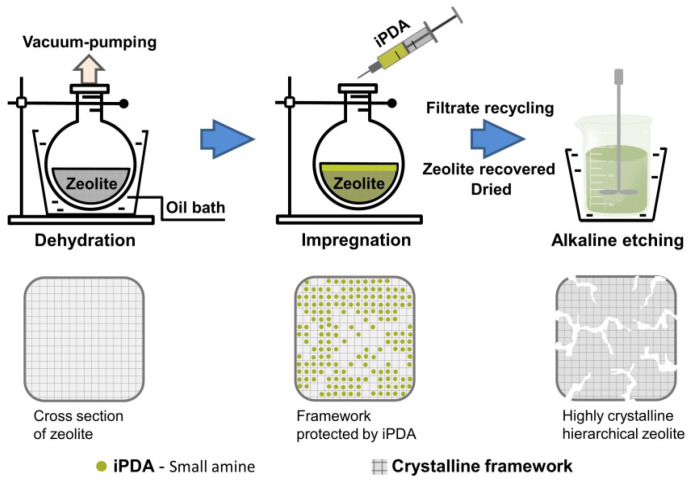
Schematic diagram of the construction of highly crystalline hierarchical zeolite with small amines as the inner pore-directing agents during the alkaline treatment. Reproduced with permission from [21], published by John Wiley and Sons, 2024.

**Figure 5 molecules-30-03798-f005:**
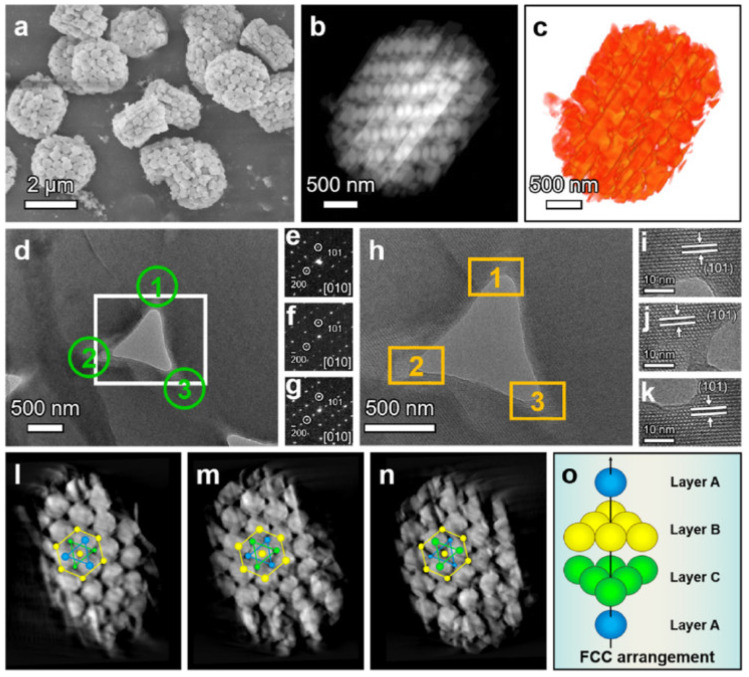
(**a**–**o**) SEM, HAADF-STEM and HR-TEM images of hierarchical ZSM-5(380, 200) zeolites. Reproduced with permission from authors of [31], published by Elsevier, 2025.

**Figure 6 molecules-30-03798-f006:**
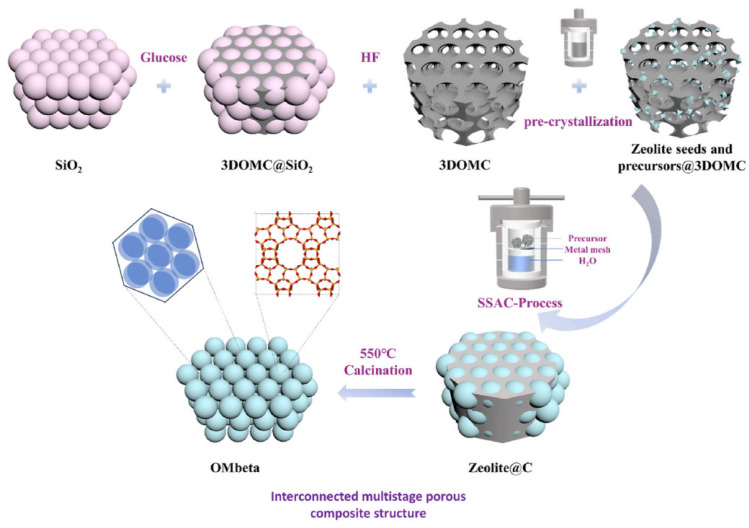
The OMbeta zeolite formation process. Reproduced with permission from authors of [3], published by Elsevier, 2025.

**Figure 7 molecules-30-03798-f007:**
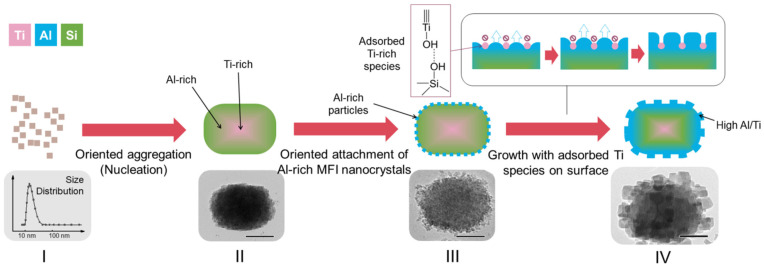
Mechanism for fractal growth of single-crystalline hierarchical zeolites with blackberry-like morphology [1].

**Figure 8 molecules-30-03798-f008:**
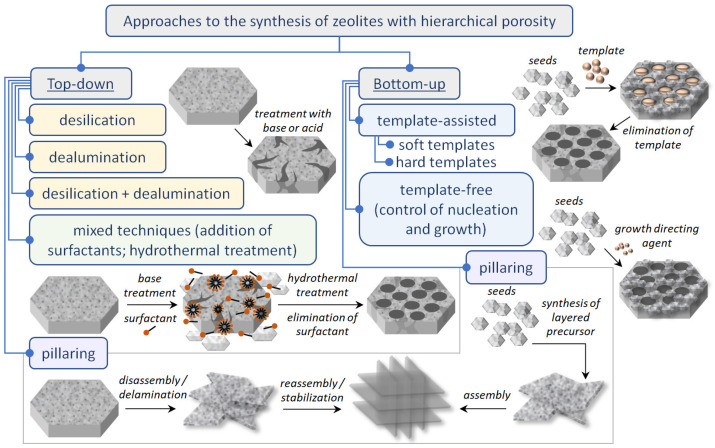
Approaches to the synthesis of hierarchical zeolites.

**Figure 9 molecules-30-03798-f009:**
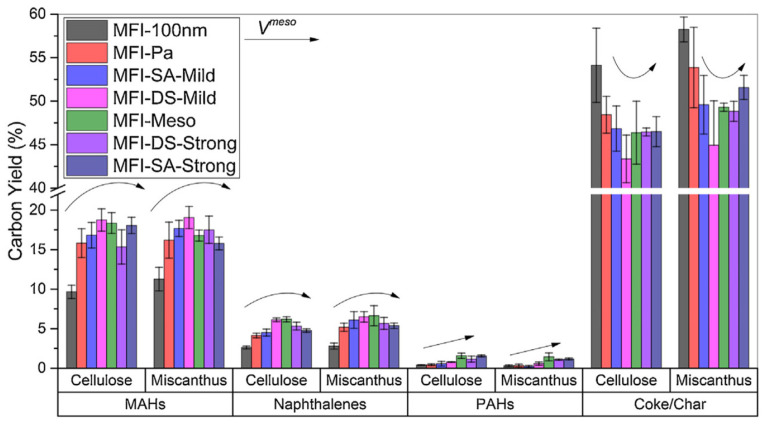
Lumped yields to MAHs (including BTX, alkyl benzenes and indenes), naphthalenes, PAHs and coke and char plotted in order of increasing mesopore volume (reaction conditions: 600 °C, residence time 20 s, loading: 5 mg total; 5 mg catalyst/mg biomass). Reproduced with permission from authors of [75], published by Elsevier, 2016.

**Figure 10 molecules-30-03798-f010:**
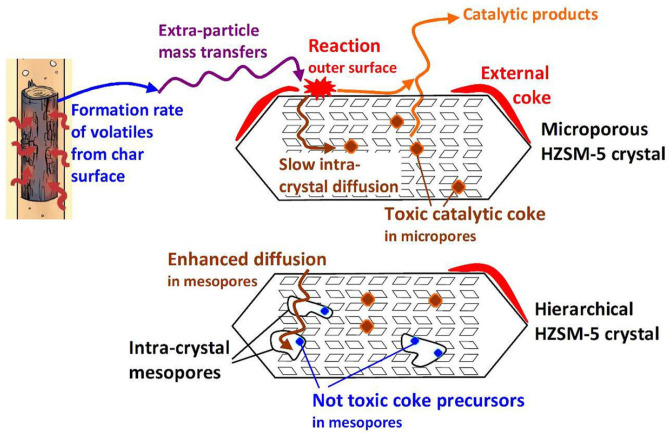
Simplified scheme highlighting the main physical–chemical mechanisms occurring during biomass CFP and different type of cokes. Reproduced with permission from [76], published by RSC, 2017.

**Figure 11 molecules-30-03798-f011:**
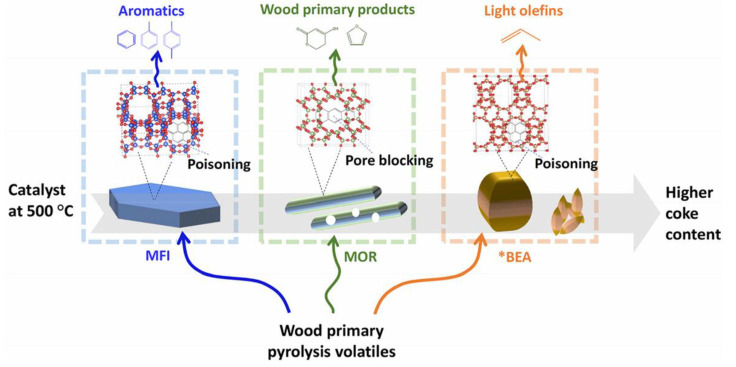
Main species produced by the three types of zeolites and effect of coke deposit on their deactivation mechanism (fixed bed, BCR 0.8) [82].

**Figure 12 molecules-30-03798-f012:**
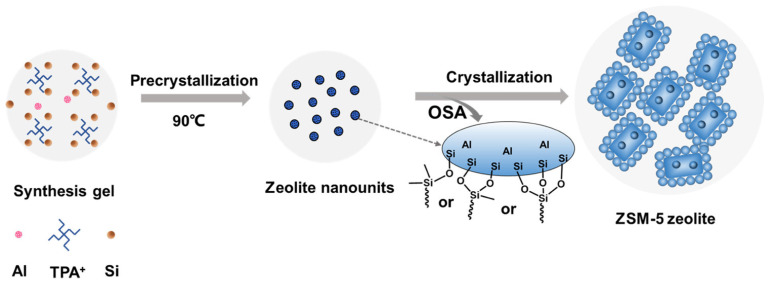
Preparation of hierarchical ZSM-5 zeolite with OSA [89].

**Figure 13 molecules-30-03798-f013:**
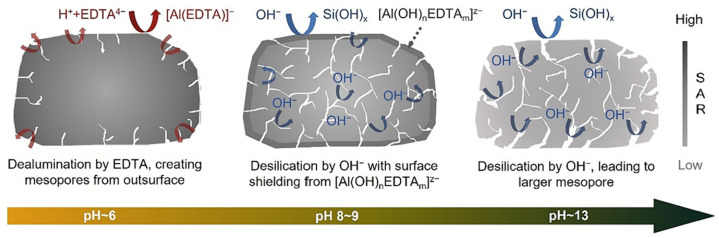
Schematic of the post-synthetic chelation–alkaline treatment of ZSM-5 zeolite at different pH values [91].

**Figure 14 molecules-30-03798-f014:**
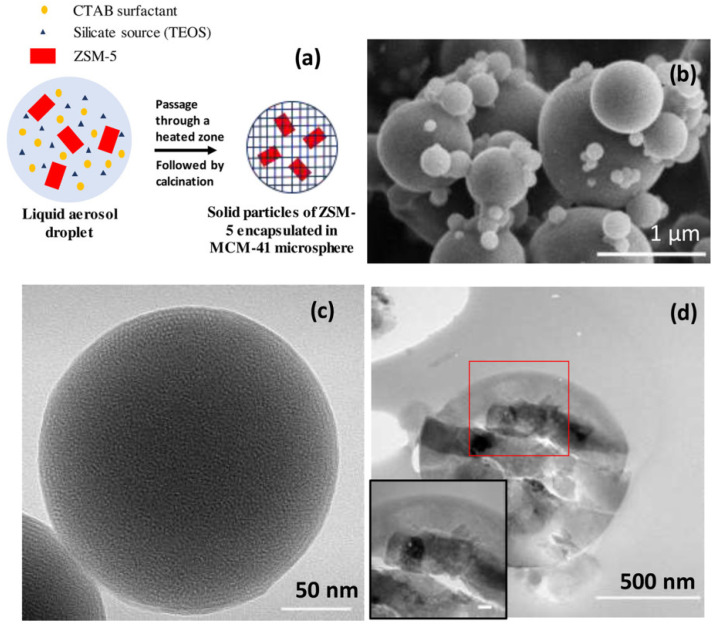
Scheme of the encapsulation of ZSM-5 in MCM-41 microspheres (**a**) and electron microscopy images of resulting particles (**b**,**c**) including the cut-section of MCM-41/50%ZSM-5 (**d**). Reproduced with permission from authors of [103], published by Elsevier, 2020.

**Figure 15 molecules-30-03798-f015:**
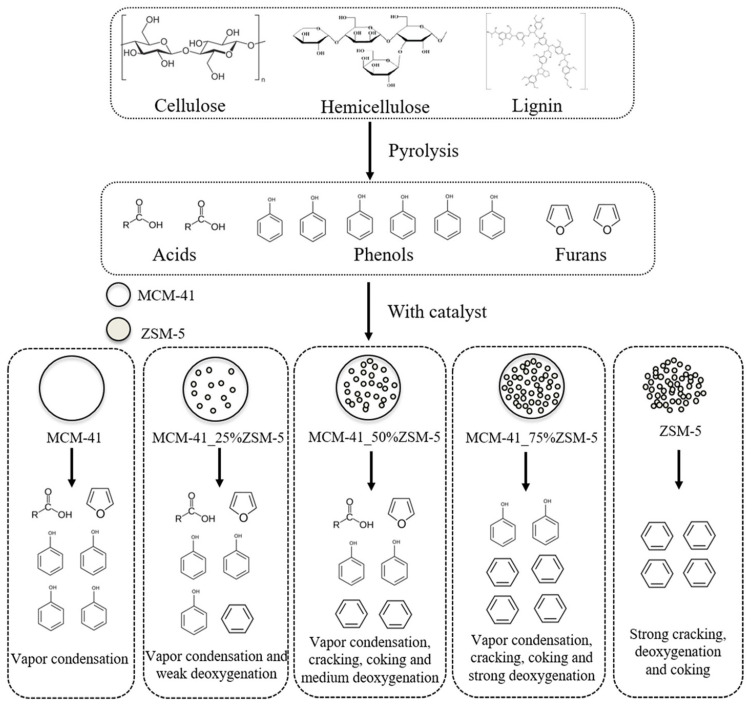
Simplified scheme of the CFP of miscanthus using the MCM-41/ZSM-5 composites. Reproduced with permission from authors of [103], published by Elsevier, 2020.

**Figure 16 molecules-30-03798-f016:**
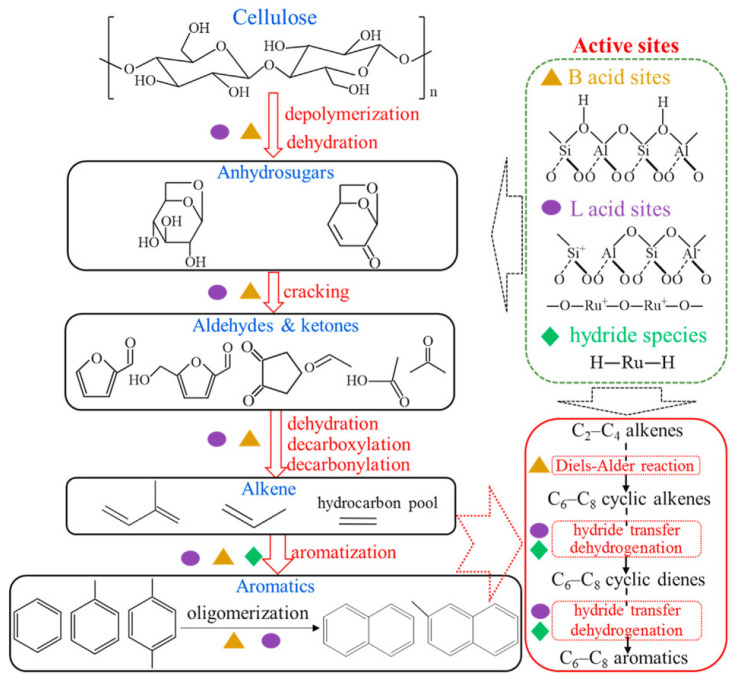
Plausible reaction mechanism for cellulose pyrolysis over xRu-MZSM. Reproduced with permission from authors of [124], published by Elsevier, 2022.

**Figure 17 molecules-30-03798-f017:**
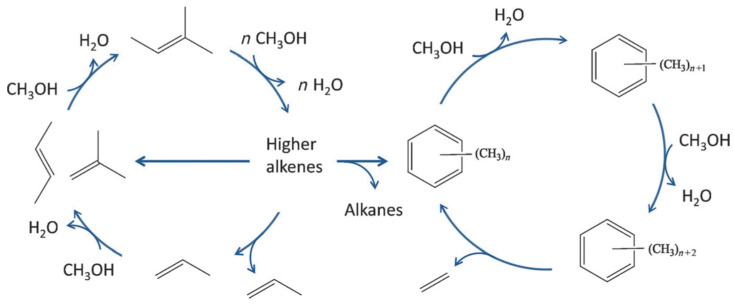
Suggested dual-cycle concept for the conversion of methanol over H-ZSM-5. Reproduced with permission from [126], published by John Wiley and Sons, 2012.

**Figure 18 molecules-30-03798-f018:**
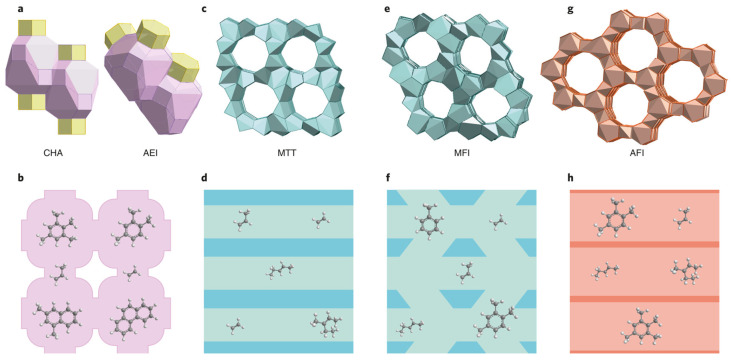
Impact of topology on cycles propagation: (**a**,**b**) 8-membered ring zeolites-aromatics can be formed, but cannot escape the cage, thus, high selectivity to short-chain olefins will be observed; (**c**,**d**) 1D 10-membered ring zeolites-cannot accommodate aromatic molecules, products are mainly gasoline range of hydrocarbons; (**e**,**f**) 3D 10-membered ring MFI structure-can host the aromatic cycles, wide range of hydrocarbons can be formed; (**g**,**h**) channels of 12-membered ring zeolites-can host aromatic hydrocarbons, which can run inside the channels. Reproduced with permission from [128], published by Springer Nature, 2018.

**Figure 19 molecules-30-03798-f019:**
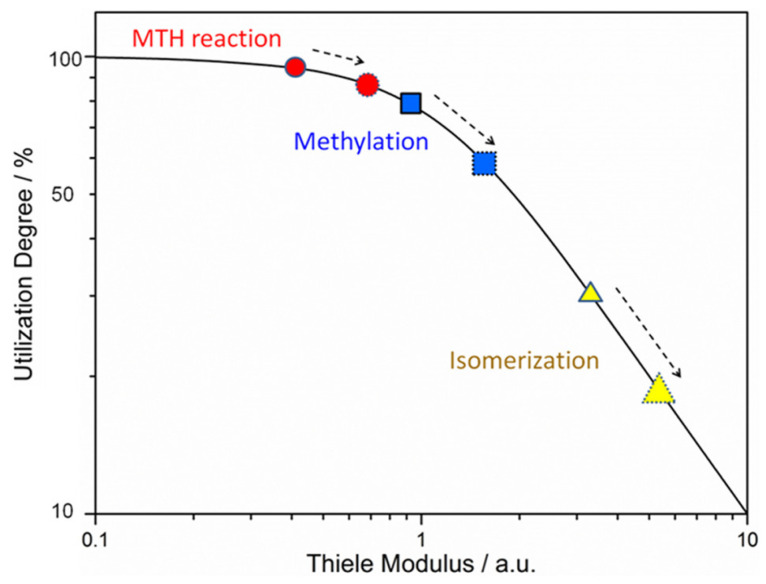
Thiele modulus and degree of utilization of ZSM5-P (scatters with solid line, crystal size 120 nm) and ZSM5 250 nm (scatters with dash line, crystal size 250 nm) in the methylation of toluene (square), MTH reaction (circle), and isomerization (triangle) at 753 K, 4.5 kPa of toluene, 1.1 kPa of MeOH [132].

**Figure 20 molecules-30-03798-f020:**
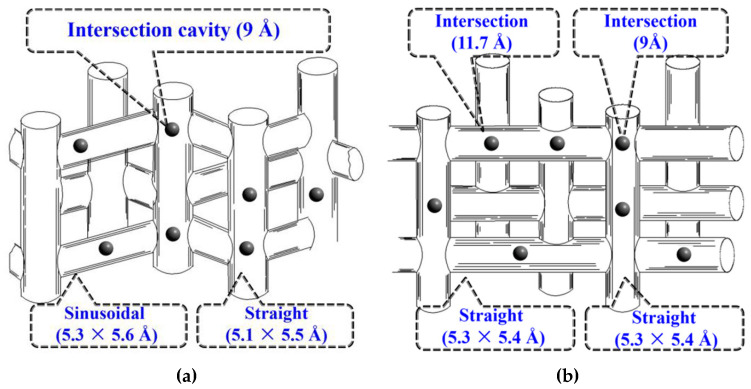
Topological structures of ZSM-5 (**a**) and ZSM-11 (**b**) zeolites with labeled pore sizes of corresponding channels [143].

**Figure 21 molecules-30-03798-f021:**
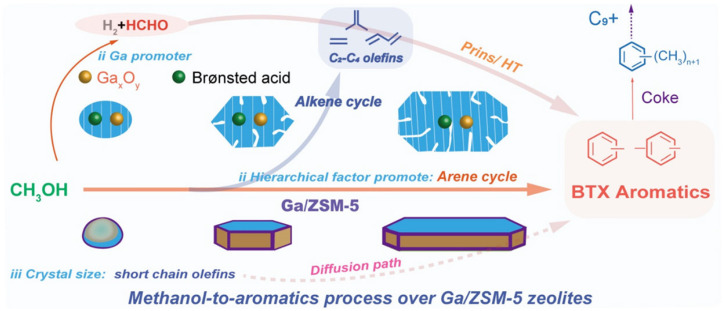
A mechanistic overview summarizing the cumulative impact of Ga promotion, particle size, and hierarchy to facilitate the aromatics yield. Reproduced with permission from [146], published by RSC, 2024.

**Figure 22 molecules-30-03798-f022:**
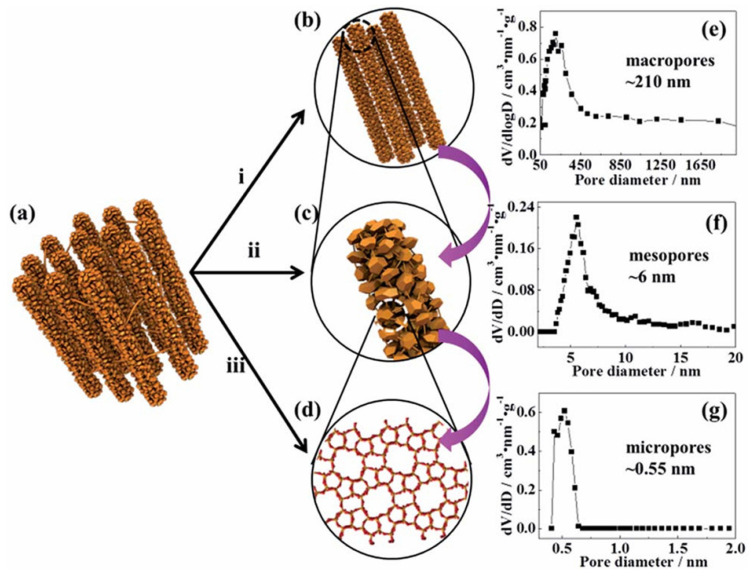
Pore properties of hierarchical MFI zeolite nanorod arrays: (**a**–**d**) schematic of the three classes of porosity in the hierarchical structure; (**e**–**g**) corresponding pore size distributions. Reproduced with permission from [150], published by RSC, 2016.

**Figure 23 molecules-30-03798-f023:**
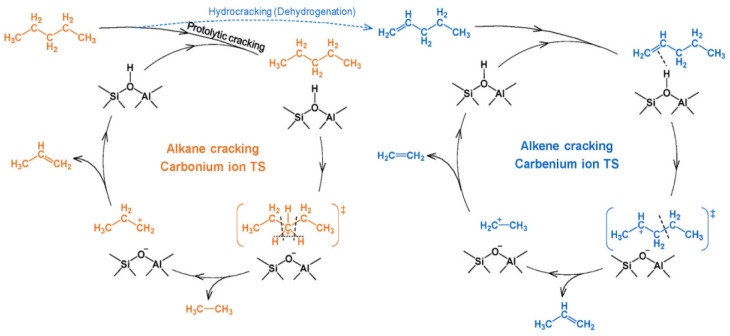
Alkane cracking via quintuple coordinated carbonium ion transition state (left, in orange) and alkene cracking via tetrahedrally coordinated carbenium ion transition state (right, in blue). Reproduced with permission from [187], published by Elsevier, 2023.

**Figure 24 molecules-30-03798-f024:**
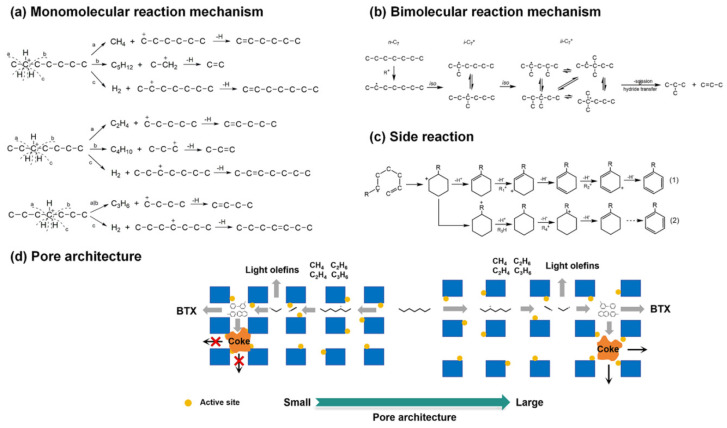
Catalytic cracking mechanism of *n*-heptane. (**a**) Monomolecular reaction mechanism, (**b**) bimolecular reaction mechanism, (**c**) side reaction, and (**d**) effect of catalyst pore architecture on catalytic properties. Reproduced with permission from authors of [178], published by Elsevier, 2024.

**Figure 25 molecules-30-03798-f025:**
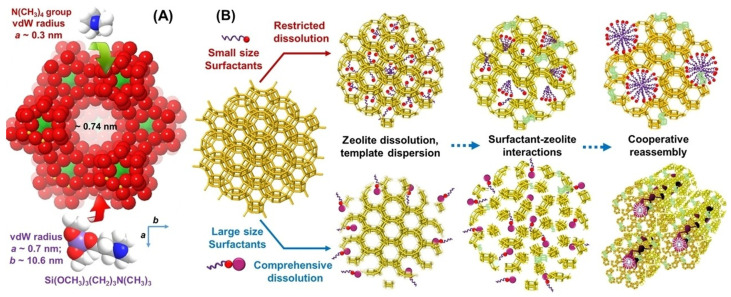
(**A**) Molecular size comparison (vdW = van der Waals radius) of surfactant head groups with FAU zeolite pore; (**B**) proposed pathways for hierarchical ordering by post-synthetic ensembles. Reproduced with permission from [180], published by John Wiley and Sons, 2024.

**Figure 26 molecules-30-03798-f026:**
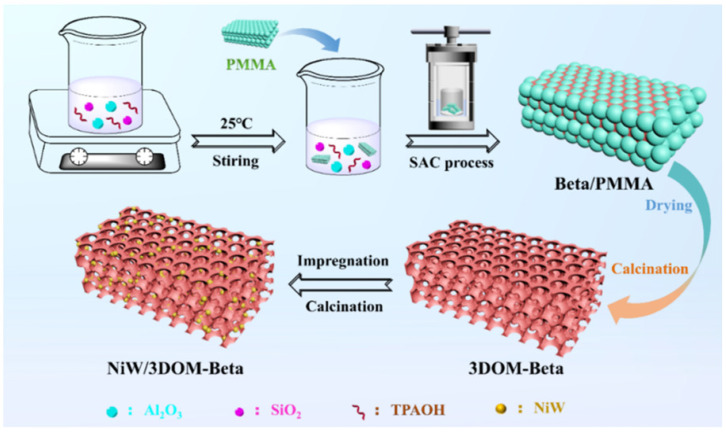
Preparation diagram of NiW/3DOM-Beta catalyst. Reproduced with permission from authors of [184], published by Elsevier, 2024.

**Figure 27 molecules-30-03798-f027:**
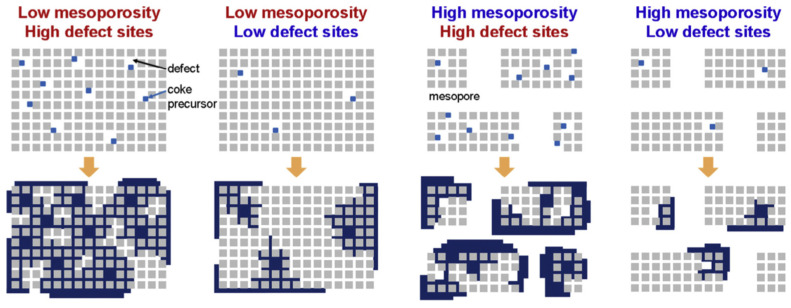
Schematic illustration of coke deposition behaviors on zeolite catalysts with different defect contents and porous structures. Reproduced with permission from [192], published by Elsevier, 2017.

**Table 1 molecules-30-03798-t001:** Application of hierarchical zeolites in pyrolysis of biomass and polymeric wastes.

Zeolite (Designation)	Synthesis Method of Hierarchical Zeolite	Feedstock	Pyrolysis Conditions	Yield or Selectivity to Aromatics	Ref.
HZSM-5 (MFI-Meso)	bottom-up (TPAOH + NaOH + hydrothermal teatment)	miscanthus	*in situ*, quartz microreactor, 600 °C, residence time (r.t.) 20 s, total loading 5 mg, BCR 0.2	yield of MAHs is 18.0%	[75]
HZSM-5 (MFI-SA-Mild)	top-down (NaOH)	miscanthus	*in situ*, quartz microreactor, 600 °C, r.t. 20 s, total loading 5 mg, BCR 0.2	yield of MAHs is 17.4%	[75]
HZSM-5 (MFI-DS-Mild)	top-down (NaOH + CTAB)	miscanthus	*in situ*, quartz microreactor, 600 °C, r.t. 20 s, total loading 5 mg, BCR 0.2	yield of MAHs is 19.0%	[75]
MCM-41_75%ZSM-5	spray drying of ZSM-5 suspension with CTAB and TEOS	miscanthus	*in situ*, quartz microreactor, 600 °C, r.t. 20 s, total loading 5 mg, BCR 0.2	yield of aromatics is ~20%	[103]
HZSM-5	top-down (NaOH)	Oak	MFBR, 500 °C, height of the fluidized bed 5 cm, BCR 0.85	yield of aromatics is 6.2%	[76]
HZSM-5 (HZ-0.01)	bottom-up (TPAOH + CTAB)	rice straw	*in situ*, tandem μ-reactor system, 600 °C, total loading 5 mg, BCR 0.05	yield of aromatics is 26.8%	[77]
HZSM-5	top-down (NaOH)	kraft lignin	*in situ*, micro-furnace system, 500 °C, kraft lignin 0.3 mg, BCR 0.1	selectivity to BTX is ~27%	[78]
Hβ	top-down (NaOH)	kraft lignin	*in situ*, micro-furnace system, 500 °C, kraft lignin 0.3 mg, BCR 0.1	selectivity to BTX is ~27%	[78]
HZSM-11 (NS-HZSM-11(20)-0.3)	bottom-up (TBAOH)	maize straw	*in situ*, fixed-bed reactor, 500 °C, maize straw 1 g, BCR 1	N/A (23.6% of bio-oil, hydrocarbons 63.6%)	[81]
HZSM-5	top-down (NaOH)	oakwood	*in situ*, double fixed bed microreactor, 773 K, oakwood 85 mg, catalyst 100 mg, BCR 0.8	yield of MAHs is ~4–5%	[82]
Beta(36) (B36-0.3-24)	top-down (TEAOH + hydrothermal teatment)	lignin	*in situ*, fixed-bed reactor, 600 °C, lignin 1.0 g, catalyst 1.0 g	yield of bio-oil is 13.1%, 56.0%MAHs	[83]
ZSM-5-F	bottom-up (TPAOH, TEAOH, HF)	stem wood	*in situ*, micro-pyrolyser, 500 °C, feedstock 0.4 mg, catalyst 2.0 mg	selectivity to BTX is 20.6%	[84]
ZSM-5 (ODDMMS (0.05)-Z5)	bottom-up (TPAOH + OSA)	cellulose	*in situ*, tandem μ-reactor system, 600 °C, cellulose 4 mg, BCR 0.05	yield of aromatics is 42.2%	[89]
CBV80-ZM (ZM is HZSM-5/MCM-41)	top-down (NaOH + CTAB)	PP	*ex situ*, micro-pyrolyzer, 550 °C, feedstock 0.4 mg, catalyst 32 mg	total light olefins and MAHs yield is 92%	[90]
ZSM-5 (Z-HT-0.3-3)	top-down (MW, Na_2_H_2_EDTA + NaOH)	HDPE	*ex situ*, two-stage pyrolysis-catalysis reactor, 500 °C, HDPE 4 g, catalyst 0.4 g	~12% yield of oil, aromatics is78.7%	[91]
HZSM-5	top-down (NaOH)	TPW + HDPE	*in situ*, 550 °C, TPW: HDPE = 1, catalyst 3 g, BCR 1	yield of MAHs is 71.75%	[92]
HZSM-5 covered with MCM-41 layer (HM-10%CTAB)	top-down (NaOH + CTAB)	bamboo	*ex situ*, Py-GC/MS, 600 °C, bamboo 1 mg, BCR 0.5	N/A (yield of hydro- carbons is 53.23%)	[94]
HZSM-5/MCM-41	top-down (NaOH + CTAB)	rice husk (R) + greenhouse plastic films (W)	*ex situ*, Py-GC/MS, 600 °C, feedstock 1 mg, R/W = 1:1.5, BCR 0.5	71.1% of hydro- carbons, >43% MAHs	[95]
HZSM-5/MCM-41	top-down (TPAOH + CTAB)	rice husk	*in situ*, MACFP, 550 °C, rice husk 8 g, BCR 1	60.5% of hydro- carbons, 43.5% MAHs	[96]

**Table 2 molecules-30-03798-t002:** Application of metal-doped hierarchical zeolites in pyrolysis of biomass.

Catalyst (Metal Content)	Synthesis Method of Hierarchical Zeolite	Feedstock	Pyrolysis Conditions	Yield or Selectivity to Aromatics	Ref.
MgO/h-ZSM-5 (8.4 wt.% of Mg)	bottom-up (TPAOH + PHAPTMS)	eucalyptus woodchips	*ex situ*, two-zone fixed-bed reactor, 500 °C, catalyst 1 g, BCR 5	aromatics is ~6% of bio-oil	[118]
MgO/h-Beta (8.7 wt.% of Mg)	bottom-up (TEAOH + PHAPTMS)	eucalyptus woodchips	*ex situ*, two-zone fixed-bed reactor) 500 °C, catalyst 1 g, BCR 5	aromatics is ~1.5% of bio-oil	[118]
ZnO/h-ZSM-5 (9.7 wt.% of Zn)	bottom-up (TPAOH + PHAPTMS (silanization agent))	eucalyptus woodchips	*ex situ*, two-zone fixed-bed reactor, 500 °C, catalyst 1 g, BCR 5	aromatics is ~5% of bio-oil	[118]
ZnO/h-Beta (10 wt.% of Zn)	bottom-up (TEAOH + PHAPTMS)	eucalyptus woodchips	*ex situ*, two-zone fixed-bed reactor, 500 °C, catalyst 1 g, BCR 5	aromatics is ~1.5% of bio-oil	[118]
La/Hi-ZSM-5 (5 wt.% of La)	top-down (Na_2_CO_3_)	rape straw	*ex situ*, two-stage fixed- bed reactor, 500 °C, rape straw 150 g, catalyst 30 g	aromatics is 49.86% of bio-oil organic fraction	[119]
Ga/HZSM-5-0.3M (1 wt.% of Ga)	top-down (NaOH)	cellulose	*in situ*, micro-pyrolyzer, 500 °C, total loading 10 mg, r.t. 30 s, BCR 0.05	selectivity to BTXis 53.70%	[120]
Fe/HZSM-5-0.3M (1 wt.% of Fe)	top-down (NaOH)	cellulose	*in situ*, micro-pyrolyzer, 500 °C, total loading 10 mg, r.t. 30 s, BCR 0.05	selectivity to BTX is 58.13%	[120]
0.1%Cu/AHZ-0.2	top-down (NaOH)	rice straw	*in situ*, tandem μ-reactor system, 600 °C, total loading 5 mg, BCR 0.05	yield of aromatics is 29.2%	[121]
0.1%Ni/AHZ-0.2	top-down (NaOH)	rice straw	*in situ*, tandem μ-reactor system, 600 °C, total loading 5 mg, BCR 0.05	yield of aromatics is 28.0%	[121]
0.5%Ga/AHZ-0.2	top-down (NaOH)	rice straw	*in situ*, tandem μ-reactor system, 600 °C, total loading 5 mg, BCR 0.05	yield of aromatics is 28.1%	[121]
8 wt.% Ni- hierarchical ZSM-5	top-down (NaOH)	torrefied corn cob	*in situ*, double-shot pyrolyzer, 550 °C, biomass 2 mg, BCR 0.5	aromatics is 54.42% of bio-oil	[122]
Fe(4)/Hie-ZSM-5 (4 wt.% of Fe)	top-down (NaOH)	poplar sawdust	*in situ*, Py-GC/MS reactor, 550 °C, total loading 0.1 g, BCR 1	aromatics is 19.92%	[123]
2Ru/MZSM (2 wt.% of Ru)	top-down (NaOH)	cellulose	*in situ*, analytical pyroprobe reactor, 650 °C, cellulose 0.2 mg, BCR 0.1	yield of aromatics is 16.8%	[124]

## Data Availability

Data sharing is not applicable to this article.

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
