# Peer review of "Enhancing the Catalytic Performance of Zeolites via Metal Doping and Porosity Control"

_molecules, 2025, doi:10.3390/molecules30183798_

Round 1

Reviewer 1 Report

Comments and Suggestions for Authors

The manuscript provides a comprehensive review of hierarchical zeolites, metal-doped ones, and their catalytic applications, particularly in fuel production and biomass conversion. The content is well-organized and covers a broad range of reactions, including pyrolysis, alcohol-to-hydrocarbon conversion, cracking, and hydroisomerization. However, several areas require clarification, expansion, or correction to enhance the manuscript's impact and readability.

  1. Abstract: The use of ‘metal-modifiers’ may not be accurate. Metals are also active sites.
  2. The introduction could better highlight the novelty of hierarchical zeolites compared to conventional zeolites. A brief comparison of hierarchical vs. traditional zeolites in terms of cost and industrial applicability would strengthen the motivation.
  3. A conceptual diagram summarizing and comparing "top-down" and "bottom-up" approaches would be valuable.
  4. Page 3, the meaning of MOR-A-AF and MOR-A could be explained.
  5. I would expect more expanded Table 1 and Table 2 considering that hierarchical zeolites and metal-doped counterparts are research hotspots.
  6. Title says ‘Metal-Doping’ but Table 2 lists many metal-oxide-doped catalysts, please clarify that.
  7. The manuscript extensively discusses pore structure and acidity but lacks mechanistic details for key reactions (e.g., MTH, hydroisomerization). Reaction pathways or schematics could be added (e.g., for MTH over ZSM-5) to aid understanding.
  8. Add a short section on limitations, challenges and future trends after the conclusions part is highly recommended.
  9. Grammar issues could be addressed, such as ‘obviousbenefit’ in the conclusion, repetitive phrases like "it is noteworthy" appears frequently
  10. Figures taken from references need to obtain permission.

Author Response

Comment 1: Abstract: The use of ‘metal-modifiers’ may not be accurate. Metals are also active sites.

 Response 1: We agree with the reviewer and clarified the sentence as “…metal-dopants (modifiers or catalysts)”

 Comment 2: The introduction could better highlight the novelty of hierarchical zeolites compared to conventional zeolites. A brief comparison of hierarchical vs. traditional zeolites in terms of cost and industrial applicability would strengthen the motivation.

 Response 2: Advantages, prospects, and some challenges related to industrial applicability of hierarchical zeolites have been highlighted in the Introduction.

Comment 3: A conceptual diagram summarizing and comparing "top-down" and "bottom-up" approaches would be valuable.

 Response 3: We agree that the data summary is useful and added a scheme (Figure 8) summarizing the approaches to the porosity formation.

Comment 4: Page 3, the meaning of MOR-A-AF and MOR-A could be explained.

 Response 4: The explanation has been added on Page 2 of the revised manuscript.

 Comment 5: I would expect more expanded Table 1 and Table 2 considering that hierarchical zeolites and metal-doped counterparts are research hotspots.

Response 5: In our review we analyzed literature for the last 10 years. That is why initially Tables 1 and 2 included only the data for this period. Nevertheless, after a careful consideration we added most important references to Section 2.1.1 and also the information to Table 1.

 Comment 6: Title says ‘Metal-Doping’ but Table 2 lists many metal-oxide-doped catalysts, please clarify that.

 Response 6: In our opinion “metal-doping” includes both doping with metals and metal oxides. The oxidation state of metals after doping is often mixed/changed so it would be difficult to give such details in the title.

Comment 7: The manuscript extensively discusses pore structure and acidity but lacks mechanistic details for key reactions (e.g., MTH, hydroisomerization). Reaction pathways or schematics could be added (e.g., for MTH over ZSM-5) to aid understanding.

 Response 7: The schemes of the reaction pathways were added on pages 17 and 23 of the revised manuscript.

 Comment 8: Add a short section on limitations, challenges and future trends after the conclusions part is highly recommended.

 Response 8: The section of “Challenges and Outlook” has been added after the Conclusion.

 Comment 9: Grammar issues could be addressed, such as ‘obvious benefit’ in the conclusion, repetitive phrases like "it is noteworthy" appears frequently.

 Response 9: Grammar was carefully checked and corrected.

 Comment 10: Figures taken from references need to obtain permission.

Response 10: For all the Figures whose reprinting requires permissions, permissions were obtained and provided to the editorial office. Note, that if only one Figure is reprinted from a reference, ACS requires no permission. Figures reprinted from the MDPI papers also need no permissions in the other MDPI journal.

Reviewer 2 Report

Comments and Suggestions for Authors

In this article, the authors reviewed the applications of  hierarchical  materials especially zeolite in catalytic cracking, hydrogenation deoxygenation, and isomerization, and provided a detailed explanation of their impact on catalytic reaction behavior. Overall, the review is comprehensive and systematical. Below are some minor revision suggestions.
1. When discussing Fig. 10, among the various molecular sieves compared, ZSM-5 demonstrated the best performance in the preparation of MAH. The author should provide an explanation for this.  
2. In the paragraph below Figure 14, the author mentions the positive effects of introducing several metals, particularly highlighting the optimal performance of mesoporous HZ40 when Mg is introduced. The possible reasons for this should be briefly explained.  
3. As a review paper, the conclusion section should not only summarize the content of the paper but also include some outlook to provide guidance or reference for future research.

Author Response

Comment 1: When discussing Fig. 10, among the various molecular sieves compared, ZSM-5 demonstrated the best performance in the preparation of MAH. The author should provide an explanation for this.

 Response 1: The explanation has been added on page 9 of the revised manuscript. The best performance is due to better shape-selectivity of ZSM-5 as compared to other zeolites.

Comment 2: In the paragraph below Figure 14, the author mentions the positive effects of introducing several metals, particularly highlighting the optimal performance of mesoporous HZ40 when Mg is introduced. The possible reasons for this should be briefly explained.

 Response 2: The explanation has been added on page 16 of the revised manuscript. This positive effect is due to the fact that the Mg-doped sample contained rather high amount of LAS and the lowest amount of BAS on the surface, resulting in higher deoxygenation rate.

Comment 3: As a review paper, the conclusion section should not only summarize the content of the paper but also include some outlook to provide guidance or reference for future research.

 Response 3: The section “Challenges and Outlook” has been added to the revised manuscript.

Reviewer 3 Report

Comments and Suggestions for Authors

Comments to the Authors:

The porosity structure and central metals of zeotype catalysts are critical factors influencing their catalytic performance. In this work, the authors summarize the strategies and methods used to control the porosity of hierarchical zeolite catalysts, as well as their applications in the production and upgrading of liquid fuels, such as in the pyrolysis of biomass and polymeric wastes. The manuscript is well written and organized. However, several issues should be addressed before publication:

1. It is recommended to include a table summarizing the strategies or methods used to control porosity.

2. The characterization techniques for analyzing porosity and center metals should be provided and comprehensively reviewed.

3. Regarding the use of zeotype materials as supports for central metals, the incorporation and/or encapsulation methods should be reviewed or summarized.

4. For the catalytic applications, the relationships between porosity structure and central metals should be highlighted and summarized.

Author Response

Comment 1: It is recommended to include a table summarizing the strategies or methods used to control porosity.

 Response 1: Figure has been added on page 6 of the revised manuscript, which summarizes the methods of creating porosity.

 Comment 2: The characterization techniques for analyzing porosity and center metals should be provided and comprehensively reviewed.

 Response 2: We believe the characterization methods for porosity and metal centers are well-known analytical techniques, described in many books and review papers. That is why we think such a description is beyond the scope of this review.

Comment 3: Regarding the use of zeotype materials as supports for central metals, the incorporation and/or encapsulation methods should be reviewed or summarized.

 Response 3: We added the general information regarding the existing methods of metal incorporation in the Introduction section.

Comment 4: For the catalytic applications, the relationships between porosity structure and central metals should be highlighted and summarized.

 Response 4: We agree with the reviewer but unfortunately, we did not find such correlations in the literature reviewed.

Reviewer 4 Report

Comments and Suggestions for Authors

In this review, the authors discussed the development of the hierachical zeolites and the metal doping on the hierachical zeolites in details. It is a nice review that shows several research aspects in zeolites, including the synthesis of hierachical zeolites, heteroatoms introduction, and catalysis study.

It would be great if the authors can write one more part in this review paper to introduce the self-pillared pentasil (SPP) zeolite publishe in Science 2012. This is another classical type of hierachical zeolites containing both micro and meso pores. Sn-SPP, W-SPP, and Pd or Pt SPP were also developed and used for various reaction. This is a big system and I recommend the authors to discuss a bit on this materials, based on their excellent performance. The finned zeolite can also be discussed together with SPP zeolites, although the study on heteroatoms substituted finned zeolites is less than that of SPP zeolites.

In conclusion, this is a nice review. I recommend it to be published in Molecules after minor revision.

Author Response

Comment 1: It would be great if the authors can write one more part in this review paper to introduce the self-pillared pentasil (SPP) zeolite publishe in Science 2012. This is another classical type of hierachical zeolites containing both micro and meso pores. Sn-SPP, W-SPP, and Pd or Pt SPP were also developed and used for various reaction. This is a big system and I recommend the authors to discuss a bit on this materials, based on their excellent performance. The finned zeolite can also be discussed together with SPP zeolites, although the study on heteroatoms substituted finned zeolites is less than that of SPP zeolites.

 Response 1: We agree that SPP is an interesting system, but these catalysts are rarely utilized in the reactions described in the review. Nevertheless, we added the relevant information to the Introduction section.

Reviewer 5 Report

Comments and Suggestions for Authors

Comments from Reviewer

Enhancing Catalytic Performance of Zeolites via Metal-Doping and the Porosity Control

The current form's presentation of methods and scientific results is satisfactory for publication in the Molecules journal. Some comments apply to the entire article. Please take this into account when making corrections. The minor and significant drawbacks to be addressed can be specified as follows:

Minor comments:

  1. Please add to the title the information that this is a review article.
  2. Fig. 1. Copyright? Permission? See also other figures.

Major comments:

  1. Introduction. I don't fully understand the vision for the Introduction. I would have expected more general divisions/pathways, especially regarding the detailed paper. The Introduction is too detailed. The authors should have shown/created diagrams collecting the types of materials, synthesis methods, metal incorporation, etc.
  2. Why did the authors omit the group of zeolites called "Organic Zeolite"?
  3. It would be interesting to add a chapter devoted to theoretical calculations - for example, the work of R.A. van Santen. Theoretical calculations have played a considerable role in attempts to synthesize new materials and have paved the way for much experimental work.

Sincerely,

    The reviewer.

Author Response

Minor comments:

Comment 1: Please add to the title the information that this is a review article.

Response 1: There is an indication that this is a review article in the file (and in publication) above the title in the line: type of article.

 Comment 2: Fig. 1. Copyright? Permission? See also other figures.

 Response 2: All the Figures requiring permissions, have permissions from corresponding publishers. Note, that if only one Figure was taken from a reference, ACS requires no permission. Figures taken from the MDPI papers also need no permissions to be reprinted in the MDPI journal.

Major comments:

 Comment 3: Introduction. I don't fully understand the vision for the Introduction. I would have expected more general divisions/pathways, especially regarding the detailed paper. The Introduction is too detailed. The authors should have shown/created diagrams collecting the types of materials, synthesis methods, metal incorporation, etc.

 Response 3: The Introduction has been revised and a Scheme was added to show more general information.

Comment 4: Why did the authors omit the group of zeolites called "Organic Zeolite"?

 Response 4: The organic zeolites are pseudo-zeolites, which are beyond the scope of this review.

 Comment 5: It would be interesting to add a chapter devoted to theoretical calculations - for example, the work of R.A. van Santen. Theoretical calculations have played a considerable role in attempts to synthesize new materials and have paved the way for much experimental work.

 Response 5: We agree DFT calculations are important in the catalyst research; however, we did not encounter any recent reports (for the last ten years) on computations of hierarchical systems. Following reviewer’s advice, however, we cited the works by R.A. van Santen in the relevant place.

Round 2

Reviewer 1 Report

Comments and Suggestions for Authors

The assertion that hierarchical zeolites are not used industrially in the introduction part is no longer accurate, given that the scale-up and application of mesostructured zeolite Y was documented ten years ago (ChemCatChem 2014, 6, 46 – 66). This point should be updated.

Author Response

Comment 1: The assertion that hierarchical zeolites are not used industrially in the introduction part is no longer accurate, given that the scale-up and application of mesostructured zeolite Y was documented ten years ago (ChemCatChem 2014, 6, 46 – 66). This point should be updated.

Response 1: We thank the reviewer for noticing this omission. We corrected this paragraph accordingly.

Reviewer 5 Report

Comments and Suggestions for Authors

My comments have been appropriately addressed in the revised manuscript.

Author Response

Comment 1. My comments have been appropriately addressed in the revised manuscript.

Response 1. We thank the reviewer for their attention to our work.